



# Sentinel-5P TROPOMI $NO_2$ retrieval: impact of version v2.2 improvements and comparisons with OMI and ground-based data

Jos van Geffen[1], Henk Eskes[1], Steven Compernolle[2], Gaia Pinardi[2], Tijl Verhoelst[2], Jean-Christopher Lambert[2], Maarten Sneep[1], Mark ter Linden[1,3], Antje Ludewig[1], K. Folkert Boersma[1,4], and J. Pepijn Veefkind[1,5]

[1]Royal Netherlands Meteorological Institute (KNMI), De Bilt, The Netherlands
[2]Belgian Institute for Space Aeronomy (BIRA-IASB), Brussels, Belgium
[3]Science and Technology Corporation (S[&]T), Delft, The Netherlands
[4]Wageningen University (WUR), Wageningen, The Netherlands
[5]Delft University of Technology (TUDelft), Delft, The Netherlands

**Correspondence:** J. van Geffen (geffen@knmi.nl)

**Abstract.**

Nitrogen dioxide ($NO_2$) is one of the main data products measured by the Tropospheric Monitoring Instrument (TROPOMI) on the Sentinel-5 Precursor (S5P) satellite, which combines a high signal-to-noise ratio with daily global coverage and high spatial resolution. TROPOMI provides a valuable source of information to monitor emissions from local sources such as power

plants, industry, cities, traffic and ships, and variability of these sources in time. Validation exercises of $NO_2$ version v1.2-v1.3 data, however, have revealed that TROPOMI's tropospheric vertical columns (VCDs) are too low by up to $50\%$ over highly polluted areas. These findings are mainly attributed to biases in the cloud pressure retrieval, the surface albedo climatology and the low resolution of the a-priori profiles derived from global simulations of the TM5-MP chemistry model.

This study describes improvements in the TROPOMI $NO_2$ retrieval leading to version v2.2, operational since 1 July 2021.

Compared to v1.x, the main changes are: (1) The $NO_2$-v2.2 data is based on version 2 level-1B (ir)radiance spectra with improved calibration, which results in a small and fairly homogeneous increase of the $NO_2$ slant columns of 3 to $4\%$, most of which ends up as a small increase of the stratospheric columns; (2) The cloud pressures are derived with a new version of the FRESCO cloud retrieval already introduced in $NO_2$-v1.4, which lead to a lowering of the cloud pressure, resulting in larger tropospheric $NO_2$ columns over polluted scenes with a small but non-zero cloud coverage; (3) For cloud-free scenes a surface

albedo correction is introduced based on the observed reflectance, which also leads to a general increase of the tropospheric $NO_2$ columns over polluted scenes of order $15\%$; (4) An outlier removal was implemented in the spectral fit, which increases the number of good quality retrievals over the South-Atlantic Anomaly region and over bright clouds where saturation may occur; (5) Snow-Ice information is now obtained from ECMWF weather data, increasing the number of valid retrievals at high latitudes.

On average the $NO_2$-v2.2 data have tropospheric VCDs that are between 10 and $40\%$ larger than the v1.x data, depending on the level of pollution and season; the largest impact is found at mid- and high-latitudes in wintertime. This has brought these tropospheric $NO_2$ closer to OMI observations. Ground-based validation shows on average an improvement of the negative bias





of the stratospheric (from $-6\%$ to $-3\%$), tropospheric (from $-32\%$ to $-23\%$) and total (from $-12\%$ to $-5\%$) columns. For individual measurement stations, however, the picture is more complicated, in particular for the tropospheric and total columns.

## 1 Introduction

The Tropospheric Monitoring Instrument (TROPOMI), launched on 13 Oct. 2017 aboard the polar orbiting Sentinel-5 Pre-
cursor (S5P) satellite, provides measurements of atmospheric trace gases (such as $NO_2$, $O_3$, $SO_2$, $HCHO$, $CH_4$, $CO$) and of cloud and aerosol properties. With these measurements TROPOMI, among others, continues the long record of satellite-based observations of global $NO_2$ concentrations.

The reason to monitor $NO_2$ worldwide is its important role in troposphere and stratosphere. Nitrogen oxides ($NO_x$), the combined name of both nitrogen dioxide ($NO_2$) and nitrogen oxide ($NO$), enter the atmosphere due to anthropogenic and
natural processes. They are essential precursors for the formation of ozone in the troposphere (Sillman et al., 1990) and they influence concentrations of OH and thereby shorten the lifetime of methane (Fuglestvedt et al., 1999). Tropospheric $NO_2$ plays a key role in air quality issues, as it directly affects human health (WHO, 2003). Stratospheric $NO_2$ is involved in photochemical reactions with ozone and thus may affect the ozone layer, either by acting as a catalyst for ozone destruction (Crutzen, 1970; Seinfeld and Pandis, 2006; Hendrick et al., 2012) or by suppressing ozone depletion (Murphy et al., 1993).

The TROPOMI $NO_2$ retrieval (van Geffen et al., 2020, 2021; Eskes et al., 2021) uses the three step approach that was introduced for the OMI $NO_2$ retrieval and named DOMINO (Boersma et al., 2007, 2011). This approach was also applied within the QA4ECV project (Boersma et al., 2018) which provided a consistent reprocessing for the $NO_2$ retrieval from measurement by OMI aboard EOS-Aura (Levelt et al., 2006, 2018), GOME-2 aboard MetOp-A (Munro et al., 2006, 2016), SCIAMACHY aboard Envisat (Bovensmann et al., 1999), and GOME aboard ERS-2 (Burrows et al., 1999).

Apart from the operational product described in this paper, several groups presented scientific TROPOMI $NO_2$ retrieval products over Europe (Liu et al., 2021), China (Liu et al., 2020) or Canada (Griffin et al., 2019). These products offer interesting comparisons with the operational product because of differences in the stratospheric estimate, albedo and cloud datasets, aerosol treatment, and a-priori profiles used. Douros et al. (2021) describe the impact of the a-priori profile and present a European product based on the operational product and profiles from the Copernicus Atmosphere Monitoring Service (CAMS)
European air-quality forecasts.

The first step in the $NO_2$ processing is a Differential Optical Absorption Spectroscopy (DOAS) retrieval to determine the slant column density (SCD), $N_s$, the total amount of $NO_2$ along the effective light path from sun through atmosphere to satellite; details of this step are described by van Geffen et al. (2020). Next, $NO_2$ vertical profile information from a chemistry transport model / data assimilation system that assimilates the SCDs – in the case of TROPOMI: TM5-MP (see Eskes et al.,
2021) – is used to determine the stratospheric vertical column density (VCD), $N_v^{\mathrm{strat}}$. Finally, the tropospheric VCD, $N_v^{\mathrm{trop}}$, is determined:

$$N_v^{\mathrm{trop}} = (N_s - N_v^{\mathrm{strat}} * M^{\mathrm{strat}})/M^{\mathrm{trop}} \tag{1}$$





where $M^{\mathrm{strat}}$ and $M^{\mathrm{trop}}$ are the stratospheric and tropospheric air-mass factors (AMFs), which depend on surface albedo, surface pressure, cloud fraction, cloud pressure, the $NO_2$ vertical profile, and the viewing geometry of the satellite ground pixel in question.

Validation with ground-based measurements and comparison with OMI measurements (e.g. Judd et al., 2020; Tack et al., 2021; Verhoelst et al., 2021; Lambert et al., 2021; Marais et al., 2021; Wang et al., 2020) shows that versions v1.2 and v1.3 of TROPOMI $NO_2$ data leads to: (a) tropospheric VCDs that are too low by $22\%$ to $37\%$ for clean and slightly polluted scenes, and up to $51\%$ over highly polluted areas, and (b) stratospheric VCDs that are too low by about $0.2 \times 10^{15}$ molec cm$^{-2}$ ($3.3\,\mu\mathrm{mol\,m}^{-2}$). At the same time, the SCDs of OMI measurements match those of TROPOMI very well, with TROPOMI's SCDs about $3\%$ higher on average due to SCD retrieval details (van Geffen et al., 2020).

An improved FRESCO cloud pressure retrieval, discussed by Eskes et al. (2021), is used in the $NO_2$ processing as v1.4 since 29 November 2020, which reduces the bias in the tropospheric VCDs of polluted scenes considerably (Eskes et al., 2021; Lambert et al., 2021; Riess et al., 2021).

This paper discusses updates in the TROPOMI $NO_2$ retrieval algorithm released as data version v2.2 on 1 July 2021 and investigates the effect on the SCD and stratospheric VCD (Sect. 3) and on the tropospheric VCD (Sect. 4). This processing includes the use of updated level-1b (ir)radiance spectra (see Sect. 2.1.3). The evaluation is based on a set of test data covering the four seasons (Sect. 2.1.2), produced partly with test version v2.1 and with final version v2.2; the difference between these two for the $NO_2$ is minor (see Sect. 3). Ground-based measurements from the test periods are used to evaluate the impact of the improvements on validation results (Sect. 5).

TROPOMI $NO_2$ level-2 data is reported in SI units, i.e. in mol m$^{-2}$; the conversion factor to the more commonly used unit molec cm$^{-2}$ is $6.022140 \times 10^{19}$ mol$^{-1}$.

## 1.1 TROPOMI $NO_2$ documentation and data versions

The standard operational TROPOMI $NO_2$ data product is described in the Algorithm Theoretical Basis Document (ATBD; van Geffen et al., 2021). The Product User Manual (PUM; Eskes et al., 2021) and the Product ReadMe File (PRF; Eskes and Eichmann, 2021) describe usage of the data and the data product versions. The most recent version of these documents can be found on http://www.tropomi.eu/data-products/nitrogen-dioxide/ and on https://sentinel.esa.int/web/sentinel/technical-guides/sentinel-5p/ (last access: 12 Oct. 2021) and include information on earlier versions.

The $NO_2$ data product is made by the so-called NLL2DP processor that provides the TROPOMI data products for which institutes in The Netherlands are responsible, and if either of these product algorithms is updated, the processor version is updated for all these products. In this paper product version numbers are given with two digits; in practice a third digit may be used to account for minor bug fixes.

The following is an overview of the $NO_2$ data versions and the version of the level-1b (ir)radiance spectra (Sect. 2.1.3) used as input for the retrieval, as well as the diagnostic data set (DDS; Sect. 2.1.2) versions; further details are given in the ATBD and PRF.





- NO$_2$-v1.2 with level-1b v1.0 is used as of 30 April 2018, the start of the publicly released data; this version replaced all older versions, which are therefore not discussed.

- NO$_2$-v1.3 with level-1b v1.0 is used as of 20 March 2019, with the same NO$_2$ algorithm as v1.2 but with an improvement in the input cloud data from FRESCO that affects the NO$_2$ VCDs of some ground pixels. As of this version the surface or cloud albedo is adjusted to ensure that the retrieved cloud fraction is within the range $[0:1]$, leading to more realistic cloud pressures; the same albedo treatment is used for the NO$_2$ cloud fraction as of v2.1 (Sect. 4.3).

- NO$_2$-v1.4 with level-1b v1.0 is used as of 29 Nov. 2020, with the same NO$_2$ algorithm as v1.2-1.3 but with an improvement in the input cloud data from FRESCO that affects the NO$_2$ VCDs of many ground pixels, as discussed by Eskes et al. (2021); see also Sect. 4.1.

- NO$_2$-v2.1 with level-1b v2.0 is used for test data DDS-2 and includes a number of improvements in the NO$_2$ retrieval discussed in this paper.

- NO$_2$-v2.2 with level-1b v2.0 and the same NO$_2$ algorithm as v2.1 is used for test data DDS-3 discussed in this paper and is operational as of 1 July 2021.

- NO$_2$-v2.3 with level-1b v2.0 contains no changes in the NO$_2$ data (other than some minor bug fixes; cf. Sect. 6.2) and is operational as of 14 Nov. 2021.

- NO$_2$-v2.4 with updated level-1b v2.0 and possibly updates in the NO$_2$ data (cf. Sect. 6) is scheduled for 2022 and will be used for a full mission reprocessing as of 30 April 2018, therewith replacing all previous versions.

Note that near-real time (NRT) data are not considered here; validation of both the off-line (OFFL) and NRT data has shown that results of the two processing chains do not differ significantly (Lambert et al., 2021).

## 2 Satellite data sources and data selection

### 2.1 TROPOMI aboard Sentinel-5 Precursor

#### 2.1.1 TROPOMI instrument

The Tropospheric Monitoring Instrument (TROPOMI; Veefkind et al., 2012), launched in October 2017 aboard ESA's Sentinel-5 Precursor (S5P) spacecraft, provides measurements in four channels (UV, visible, NIR and SWIR) of various trace gas concentrations, as well as cloud and aerosol properties, from an ascending sun-synchronous polar orbit, with an equator crossing at about 13:30 local time. NO$_2$ retrieval is performed from the visible band ($400 - 496$ nm), which has spectral resolution and sampling of $0.54$ nm and $0.20$ nm, with a signal-to-noise ratio of around 1500.

Individual ground pixels are $7.2$ km ($5.6$ km as of 6 Aug. 2019) in the along-track and $3.6$ km in the across-track direction at the middle of the swath. The full swath width is about $2600$ km, with which TROPOMI achieves global coverage each day,





**Table 1.** Overview of the diagnostic data set (DDS) periods processed for evaluation of the updated $NO_2$ data. Columns 3 and 4 give the start of the data that was processed. Columns 5 and 6 give the start of the data that is used for the analysis of the vertical column density (VCD), i.e. after the spin-up period needed by TM5-MP. Columns 7 and 8 give the end of the data that was processed. Note that the orbit at the start of a period may have a sensing start time just before midnight preceding the given date. The last two columns give the version number of the publicly released off-line (OFFL) and the DDS data.

| | | Start of data period | | Start of VCD period | | End of data period | | Data version | |
|---|---|---|---|---|---|---|---|---|---|
| DDS | Season | date | orbit | date | orbit | date | orbit | OFFL | DDS |
| 2 | Summer 2018 | 2018-06-25 | 03612 | 2018-06-30 | 03683 | 2018-07-06 | 03782 | v1.2 | v2.1 |
| 2 | Winter 2019 | 2018-12-25 | 06208 | 2018-12-30 | 06280 | 2019-01-05 | 06378 | v1.2 | v2.1 |
| 2 | Spring 2019 | 2019-03-25 | 07486 | 2019-03-30 | 07556 | 2019-04-05 | 07655 | v1.3 | v2.1 |
| 2 | Autumn 2019 | 2019-09-12 | 09911 | 2019-09-17 | 09982 | 2019-09-23 | 10081 | v1.3 | v2.1 |
| 3 | Autumn 2020 | 2020-09-24 | 15274 | 2020-09-29 | 15345 | 2020-10-07 | 15473 | v1.3 | v2.2 |
| 3 | 04 Apr 2019 | 2019-04-03 | 07627 | N/A | N/A | 2019-04-05 | 07647 | v1.3 | v2.2 |

except for narrow strips between orbits of about $0.5°$ wide at the equator. The swath is across-track divided in 450 ground pixels (rows) and their size remains more or less constant towards the edges of the swath (the largest pixels are $\sim 14\,\mathrm{km}$ wide).

### 2.1.2 TROPOMI observations used in this study

In order to test the $NO_2$ algorithm updates and their impact on the retrieval results, a diagnostic data set was made. DDS-2
(generated in Sept. 2020) consists for $NO_2$ of four periods of 12 days made with test processor version v2.1, and DDS-3 (generated in April 2021) consists of one period of 14 days made with final processor version v2.2; see in Table 1.

To be able to evaluate the new tropospheric and stratospheric vertical columns (VCDs), the full DDS periods are passed through the TM5-MP data assimilation system, starting from v1.x $NO_2$ fields of the day prior to the first day of the DDS periods, which means that TM5-MP needs a few days to adjust ("spin-up") to the new v2.x data. Hence, for analysis of the
10 DDS VCDs (Sect. 4) the first 5 days of each period are skipped, whereas for the analysis of the SCDs (Sect. 3) the full periods is used.

DDS-3 also contains three periods of about one day, one of which (04 Apr 2019) overlaps with one of the DDS-2 periods and is therefore included in Table 1 as it can be used to check the effect on the $NO_2$ SCD retrieval results of changes in the level-1b (ir)radiance spectra between DDS-2 and DDS-3 (Sect. 3.3).

### 2.1.3 Updates in level-1b (ir)radiance spectra

The $NO_2$ data products of versions v1.x use as input v1.0 level-1b (ir)radiance spectra. As of the switch to v2.2 of the data (cf. Sect. 1), updated v2.0 level-1b (ir)radiance spectra are used. For the DDS processing, the input also consists of v2.0 level-1b spectra.



**Table 2.** Configuration parameters in the $NO_2$ processing related to saturation in the level-1b radiance spectra and removal of outliers in the $NO_2$ retrieval residual for different versions of the $NO_2$ data, with their respective level-1b spectra version.

| Configuration parameter | $NO_2$ v1.2 – v1.4 Level-1b v1.0 | $NO_2$ v2.1 – v2.2 Level-1b v2.0 | $NO_2$ v2.1_test Level-1b v1.0 |
|---|---|---|---|
| The maximum fraction of the radiance spectrum that is allowed to be flagged as saturated before the ground pixel is skipped | 0.01 | 0.25 | 0.12 |
| The maximum number of outliers that is allowed to be in a radiance spectrum before the ground pixel is skipped | N/A | 10 | 15 |

The pre-launch calibration results, used for most of the v1.0 level-1b spectra, are described by Kleipool et al. (2018), while the updates in the level-1b spectra are detailed by Ludewig et al. (2020); see also the TROPOMI reflectance validation study of Tilstra et al. (2020). The updates most relevant for $NO_2$ are mentioned here, while Sect. 3.3 discusses the impact of the v2.0 level-1b spectra on the $NO_2$ retrieval.

Saturation effects may occur in the detectors of band 4 (visible, e.g. used for $NO_2$ retrieval) and band 6 (NIR, e.g. used for cloud data retrieval) over very bright scenes, such as complexes of high clouds, which result in lower-than-expected radiances for certain spectral (i.e. wavelength) pixels. In addition, large saturation effects may lead to so-called blooming: excess charge flows from saturated into neighbouring detector (ground) pixels in the row direction, resulting in higher-than-expected radiances for certain spectral pixels (Ludewig et al., 2020). Level-1b v1.0 spectra contain flagging for saturation but not for

blooming. Level-1b v2.0 also has flagging for blooming (Ludewig et al., 2020), where one error flag number is used for both saturation and blooming. Also improved in v2.0 spectra is flagging for transients, caused by charged particles hitting the detector, relevant all over the world, but in particular over the South Atlantic Anomaly (cf. Sect. 3.2).

Further improvements lie in the degradation correction of the irradiance, in corrections for the absolute and relative (ir)radiances, in the noise and error estimate of the irradiance spectra, and in the determination of the measurement quality (Ludewig et al.,

2020). A change in the absolute reflectance, the ratio between the radiance and irradiance, does not affect the retrieved SCDs but it has an impact on the scene albedo and cloud fraction, and therefore on the AMFs and VCDs.

In the time between the generation of DDS-2 and DDS-3 the calibration key data (CKD) of the Level-1b v2.0 spectra, including the irradiance degradation correction, were recalculated using fits over more data (for DDS-2 up to May 2019, for DDS-3 up to Feb. 2021; over the latter period the irradiance degradation was about $2.6\%$ in band 4 and less than $0.5\%$ in

band 6). This recalculation leads to minor differences for overlapping data periods of DDS-2 and DDS-3: for band 4 both radiance and irradiance differ by less than $0.1\%$. The impact on the $NO_2$ SCD retrieval results (Sect. 3.3) is negligible and is therefore not discussed here.

Implementation of a separate radiance degradation correction is at the time of writing under discussion, with the possible intent to include this in the planned mission reprocessing (cf. Sect. 6.3).





## 2.2 OMI aboard EOS-Aura

### 2.2.1 OMI instrument

The Ozone Monitoring Instrument (OMI; Levelt et al., 2006), launched in July 2004 aboard NASA's EOS-Aura spacecraft, provides measurements in three channels (two UV and one visible) of various trace gas concentrations, as well as cloud and

aerosol properties, from an ascending sun-synchronous polar orbit, with an equator crossing at about 13:40 local time. $NO_2$ retrieval is performed from the visible band ($349-504\,nm$), which has spectral resolution and sampling of $0.63\,nm$ and $0.21\,nm$, with a signal-to-noise ratio of around $500$.

Individual ground pixels are $13\,km$ in the along-track and $24\,km$ in the across-track direction at the middle of the swath. The full swath width is about $2600\,km$ and with that OMI achieves global coverage each day. The swath is across-track divided in

60 ground pixels (rows) and their size increases towards the edges of the swath to $\sim 150\,km$.

### 2.2.2 OMI observations used in this study

Comparisons of the magnitude of the TROPOMI and OMI $NO_2$ column data are done using OMI orbits from the DDS periods (Table 1) processed within the framework of the QA4ECV project (Boersma et al., 2018); validation of that data is discussed by Compernolle et al. (2020) and Pinardi et al. (2020).

Since June 2007 a part of the OMI detector suffers from a so-called row anomaly, which appears as a signal suppression in the level-1b radiance data at all wavelengths (Schenkeveld et al., 2017), leading e.g. to large uncertainties on the $NO_2$ data in the affected rows $22-53$ (0-based), so that effectively the data of these rows have to be skipped from the $NO_2$ analysis.

Due to this issue and the fact that the TROPOMI and OMI orbits do not exactly overlap, because they measure from slightly different altitudes, direct orbit-to-orbit comparisons are not possible. Instead, data comparisons in this paper are performed

after conversion to a common longitude-latitude grid.

## 3 Updates in the SCD retrieval step

### 3.1 Fit window wavelength assignment

The first step in the data processing chain is the selection of the spectral index range $[i_b : i_e]$ that comprises the wavelength window $[\lambda_b : \lambda_e]$ needed for the wavelength calibration and DOAS retrieval steps; for the $NO_2$ SCD retrieval $\lambda_b = 405\,nm$ and

$\lambda_e = 465\,nm$. The selection is done at the nominal wavelength grid assigned to the level-1b (ir)radiance spectra. For a given spectral index, $i$, the radiance wavelengths varies across the detector rows, as illustrated in Fig. 1; this is the so-called spectral smile. Consequently, each detector row has its own $[i_b : i_e]$. With $\lambda_b = 405\,nm$, for example, the level-1b v2.0 radiance nominal wavelength gives $i_b = 36$ for the rows along the swath edge and $i_b = 24$ for the central rows. Some of the rows around changes in the $i_b$ and/or $i_e$ have in v1.2-v1.4 a slightly higher SCD error estimate than neighbouring rows. This difference, which is

less than $1\,\mu mol\,m^{-2}$, is reduced by two small corrections in the spectral index selection, with little to no effect on other rows.





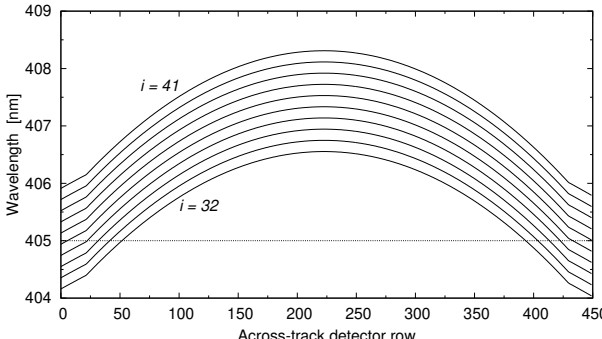

**Figure 1.** Across-track nominal wavelengths of the level-1b radiance spectra of v2.0 for a selected number of spectral indices: $i = 32$ to $i = 41$ of band 4. The horizontal dotted line marks $405\,\mathrm{nm}$.

## 3.2 Outlier removal

Spectral pixels flagged in the level-1b v1.0 data as suffering from saturation or transients (or other errors) are skipped from the measurement before the spectra are used in the further data processing.

Level-1b v1.0 spectra have no flagging for spectral pixels suffering from blooming (cf. Sect. 2.1.3), hence there may be

many problematic spectral pixels around spectral pixels that are flagged as suffering from saturation. These spectral pixels suffering from blooming have radiance levels very different from what is expected, leading to outliers (spikes) in the DOAS fit residual, which is the difference between the measured and the DOAS modelled reflectance. Similarly, level-1b data is flagged for transients caused by charged particles hitting the detector, but not all such events constitute transients and perhaps not all transient events are captured, thus leading to possible outliers in the residual.

Since the $NO_2$ v1.2-v1.4 processor does not have an algorithm that removes the spectral pixels that show such an outlier ("outlier removal") from the DOAS fit, the maximum fraction of the spectral pixels within the $NO_2$ fit window ($405 - 465\,\mathrm{nm}$, which covers 304 or 305 spectral pixels) allowed with saturation flag without skipping the ground pixels was necessarily low (Table 2). With the introduction of an outlier removal routine in $NO_2$ v2.1 (as announced by van Geffen et al., 2020; see also van Geffen et al., 2021, App. F), and the fact that Level-1b v2.0 flags the spectral pixels suffering from blooming the same way

as saturated pixels (Sect. 2.1.3), a larger fraction of the spectral pixels is allowed to be flagged as saturated (third column in Table 2). In case of outliers in the residual of a given ground pixel caused by charged particles hitting the detector, it appears that the number of spectral pixels showing outliers is usually small (less than 5), while in the case of saturation/blooming the number of outliers may be much higher. If the number of outliers is really high, the outlier removal routine may not work well, because it is applied only once (van Geffen et al., 2021), the maximum number of allowed outliers is for the operational

processing set to 10 (Table 2).

Fig. 2 shows a map of the difference "v2.1 minus v1.2" (referring to the settings listed in Table 2) in the $NO_2$ SCD error estimate from the DOAS fit for an orbit over the South Atlantic Anomaly (SAA) using the level-1b v2.0 (ir)radiance spectra as input. There are a few along-track lines visible in Fig. 2: these are rows for which most pixels have exactly one outlier at




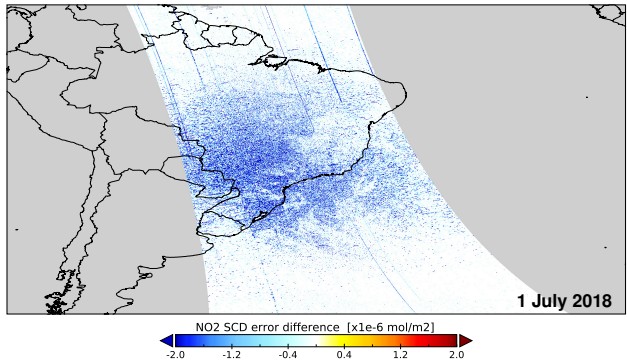

**Figure 2.** Map of the TROPOMI $NO_2$ SCD error difference of orbit 03707 of 1 July 2018 over the South Atlantic Anomaly (SAA) due to outlier removal, i.e. the SCD error using the v2.1 settings minus those using the v1.2 settings listed in Table 2, both with level-1b v2.0 spectra as input. The depicted area is longitude $= [-80° : -10°]$, latitude $= [+50° : +10°]$.

the beginning of the $NO_2$ fit window and are thus related to details of the wavelength assignment (Sect. 3.1) in the pre-v2.1 processor.

Fig. 3 shows for the same orbit along-track averages over all 633 scanlines that have a nadir latitude within $40°$ South and North of the equator. For the Southern area outlier removal with the v2.1 settings (blue line) clearly leads to a lower SCD error

than the v1.2 settings (red line), both made using the level-1b v2.0 spectra. With level-1b v1.0 spectra the SCD error (dotted gray line) is slightly higher than with the level-1b v2.0 spectra, partly because the latter has improved flagging (Sect. 2.1.3). For the Northern area outlier removal has little effect on the SCD error: there is no clear difference between the green line (v1.2 settings) and the black line (v2.1 settings), again both made using the level-1b v2.0 spectra. The red and green lines, with the v1.2 settings, show a few peaks due to some strong outliers in the residuals which are not removed. The "jumps" visible at

rows 21-22 and 429-430 are caused by changes in the on-board across-track binning of data. For the Southern (Northern) area the outlier removal removes along a given row outliers every 2-3 (5-10) scanlines.

The SCD depends strongly on the along-track and across-track variation in the solar and viewing zenith angles. To ease evaluation of the SCD, consider what could be called the geometric column density (GCD), defined as $N_v^{geo} = N_s/M^{geo}$ with $M^{geo}$ the geometric AMF, which depends only on the viewing angles. Outlier removal does not change the $NO_2$ SCD itself

significantly: along-track averages of the GCD differ by $< 0.4\,\mu mol\,m^{-2}$, except for pixels where (strong) outliers are removed; the average GCD for the Southern and Northern area are $\sim 50$ and $\sim 40\,\mu mol\,m^{-2}$, respectively.

The $NO_2$ v2.1 processor has been used with level-1b v1.0 spectra to provide dedicated $NO_2$ data files for lightning $NO_x$ studies that look at the production of $NO_2$ above bright storm clouds, where saturation/blooming may be a big issue. To this end special configuration settings, listed in the fourth column of Table 2 as "v2.1_test", are used: more outliers are accepted,

but the number of spectral pixels flagged for saturation is limited somewhat because level-1b v1.0 spectra lack flagging for blooming. With these special settings, more ground pixels can be used – but with great care – for such lightning $NO_x$ studies (Allen et al., 2021; Perez-Invernon et al., 2021; Zhang et al., 2021).





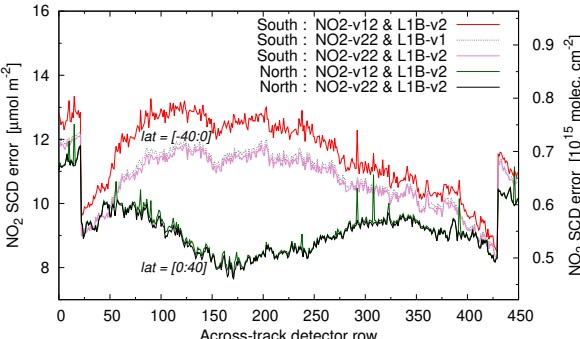

**Figure 3.** Along track averages over the latitude ranges $[0° : \pm 40°]$ of the SCD error of orbit 03707 of 1 July 2018 as function of across-track detector row number. The legend refers to the $NO_2$ processor settings for outlier removal listed in Table 2 and the version of the level-1b spectra used as input. See the text for further discussion and details.

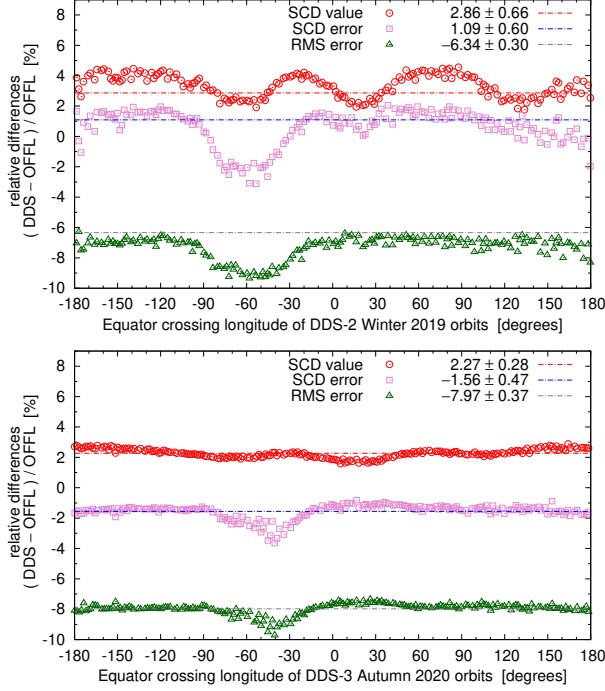

**Figure 4.** Relative differences in the DOAS retrieval results between the TROPOMI DDS and OFFL data, averaged over the $30°$ tropical latitude region during the full DDS-2 Winter 2019 (top) and DDS-3 Autumn 2020 (bottom) periods (cf. Table 1), as function of the equator crossing longitude of the orbits. The overall averages are listed in the legend and in Table 3.





**Table 3.** Relative differences in the DOAS retrieval results between the DDS and OFFL data averaged over the full DDS-2 and DDS-3 periods as well as the relative and absolute differences in the stratospheric VCD averaged over the VCD periods (cf. Table 1), for the ground pixels with valid SCD retrieval results in the the $30°$ tropical latitude region.

| | | Relative change ( DDS − OFFL ) / OFFL [%] | | | | Change in $N_v^{strat}$ | |
| | | | | corrected | | relative | absolute |
| DDS | Season | SCD value | SCD error | SCD error | RMS error | [%] | [$\mu mol\,m^{-2}$] |
|---|---|---|---|---|---|---|---|
| 2 | Summer 2018 | $2.86 \pm 0.66$ | $1.09 \pm 0.55$ | $-2.45 \pm 0.55$ | $-6.34 \pm 0.30$ | $3.18 \pm 0.37$ | $1.38 \pm 0.16$ |
| 2 | Winter 2019 | $3.32 \pm 0.74$ | $0.56 \pm 1.21$ | $-2.98 \pm 1.21$ | $-7.30 \pm 0.74$ | $3.46 \pm 0.46$ | $1.23 \pm 0.17$ |
| 2 | Spring 2019 | $3.75 \pm 0.72$ | $1.12 \pm 0.64$ | $-2.42 \pm 0.64$ | $-7.08 \pm 0.41$ | $4.36 \pm 0.36$ | $1.38 \pm 0.11$ |
| 2 | Autumn 2019 | $3.42 \pm 0.83$ | $1.70 \pm 0.37$ | $-1.84 \pm 0.37$ | $-7.14 \pm 0.28$ | $4.13 \pm 0.33$ | $1.29 \pm 0.11$ |
| 3 | Autumn 2020 | $2.27 \pm 0.28$ | $-1.56 \pm 0.47$ | $-1.56 \pm 0.47$ | $-7.97 \pm 0.37$ | $2.93 \pm 0.14$ | $1.04 \pm 0.07$ |
| 3 | 04 Apr 2019 | $2.63 \pm 0.30$ | $-2.36 \pm 0.44$ | $-2.36 \pm 0.44$ | $-7.22 \pm 0.43$ | — | — |

## 3.3 Impact on SCD retrieval results

Fig 4 shows the relative changes in the main SCD retrieval results for the DDS-2 Winter 2019 (top) and DDS-3 Autumn 2020 (bottom) periods based on averages over the tropical latitude (TL) region used by van Geffen et al. (2020) for the evaluation of the SCD uncertainties: all scanlines with sub-satellite latitude point – corresponding to the nadir viewing detector rows –
within a $30°$ latitude range that moves along with the seasons, in an attempt to filter out most of the seasonality in the $NO_2$ columns. Table 3 lists the relative changes averaged over all orbits of each of the DDS-2 and DDS-3 periods. These averages are not an exact measure but are a good indicator of the combined impact on SCD retrieval results of the above mentioned improvements and of the use of level-1b v2.0 spectra combined. Based on the evaluation of only 12 test orbits with the v1.2 $NO_2$ retrieval, van Geffen et al. (2020) estimated that the update of the level-1b (ir)radiance spectra has a small impact on the
$NO_2$ SCD value, SCD error and RMS error of on average $+2\%$, $-1\%$ and $-6\%$, respectively.

Fig. 4 shows that the impact of the new outlier removal over the SAA, discussed in Sect. 3.2, leads to a strong decrease in the SCD error and RMS error around $30 - 60°$ W, in particular in the Winter period (top) when the TL region lies just South of the equator and thus covers a large part of the SAA; in the Autumn period (bottom) the TL region lies around the equator and overlaps less with the SAA, so that for this period the TL region average changes of the error terms are smaller. This SAA
"dip" in the SCD error and RMS error leads to a somewhat larger standard deviation of the Winter 2019 overall results listed in Table 3.

What stands out from comparing the two panels in Fig. 4 and the numbers given in Table 3 is that the change of the SCD error is very different for DDS-2 and DDS-3: in DDS-2 there is a small increase, while in DDS-3 there is a stronger decrease of the SCD error. The reason for this difference is an unfortunate bug introduced in the v2.1 processor used for DDS-2 that
was repaired again in v2.2 used for DDS-3: in v2.1 there is a mistake in the calculation of the noise on the reflectance (from







**Figure 5.** Maps of gridded data averaged over the Spring 2019 VCD period (cf. Table 1) of all ground pixels with valid retrieval: the $NO_2$ GCD (top row), stratospheric VCD (middle row) and tropospheric VCD (bottom row) values of the v2.1 data (left column) and the "DDS minus OFFL" difference (right column), all in μmol m$^{-2}$. The depicted area is longitude = $[-15° : +55°]$, latitude = $[+27.5° : +62.5°]$.

the noise on the (ir)radiance spectra) and this reflectance noise determines in part (i.e. scales) the magnitude of the SCD error, as well as the $\chi^2$ of the DOAS fit (details of the DOAS fit approach are given by van Geffen et al., 2020).

Averaging the SCD error changes of the overlapping 21 orbits of the 04 Apr 2019 test data shows a clear decrease of about 3.5 % from DDS-2 to DDS-3. Using this to correct the DDS-2 SCD error differences to the DDS-3 level leads to the numbers

5    in the 5th column of Table 3: after the correction all test data periods show more or less the same decrease of the SCD error,





with some variation between the periods likely caused by differences in atmospheric circumstances and remaining seasonal effects, despite the use of a moving TL region for the averaging.

The RMS error is not affected by the reflectance noise and the numbers given in Table 3 do not show a clear difference between DDS-2 and DDS-3 (averaged over the 04 Apr 2019 period, the RMS error of DDS-3 is $0.2\%$ lower than of DDS-3),

indicating that the quality of the $NO_2$ SCD fit has not been affected by the unfortunate bug. All DDS-2 periods have comparable RMS error decreases, with possibly a somewhat larger decrease in the Autumn 2020 DDS-3 period, which may be due to a small change in the level-1b irradiance degradation correction, but may also be due to atmospheric circumstances.

The SCD values themselves show an increase of $3-4\%$ for DDS-2 and about $2.5\%$ for DDS-3, while averaged over the 04 Apr 2019 test data the DDS-3 SCD values are $1.1\%$ lower than those of DDS-2. Again the difference between DDS-2 and

DDS-3 may be due to the small change in the irradiance degradation correction (reflectances have changed by less than $0.5\%$) and/or to atmospheric circumstances.

In summary the v2.1-v2.2 DDS data, compared to the v1.2-v1.3 OFFL data, shows a much improved DOAS fit quality, a reduced SCD error, and a small increase of the SCD values (Table 3). The SCD increase shows in the top panel of Fig. 4 some East-West variation, while in the bottom panel there is hardly any such variation. On the whole, it appears that the SCD increase

is more or less uniform across the world, with little or no hotspots. Due to the physics of the subsequent $NO_2$ data assimilation, the more or less homogeneous SCD increase leads to an increase of the stratospheric $NO_2$ vertical column ($N_v^{strat}$).

The data assimilation is set up in such a way that the total column is made consistent with the TROPOMI observations over regions with small levels of air pollution (oceans, remote land regions), basically by adjusting the stratospheric column because of the minor contribution of the troposphere in those locations. A uniform increase of the TROPOMI total column will

therefore lead to a similar increase of the stratospheric vertical column, while the tropospheric columns will be hardly affected.

The two right-most columns in Table 3 list the relative and absolute differences of the $N_v^{strat}$ averaged over the TL region using the orbits of the VCD period given in Table 1. For the DDS-3 period the increase of $N_v^{strat}$ is somewhat less than for the four DDS-2 periods, like with the SCD values and for the same reasons.

Fig. 5 shows as example maps over Europe of selected datasets gridded on $0.8° \times 0.4°$ and averaged over the VCD part

of the Spring 2019 period (cf. Table 1) using all pixels with a valid retrieval (`qa_value` $> 0.50$). The top row shows on the left the GCD of v2.1, with clear hotspots over polluted areas, while the "DDS minus OFFL" difference on the right is fairly homogeneous. The second row shows on the left the stratospheric VCD of v2.1 and on the right the difference between the v2.1 and v1.3 data, which features a more or less homogeneous increase of the stratospheric VCD, although over some strongly polluted areas, such as Moscow and St. Petersburg, the stratospheric VCD change can be larger. As a reference, the bottom row

shows the corresponding tropospheric VCD v2.1 value (left) and difference (right), with the latter also influenced by changes in the AMF discussed in Sect. 4.

Fig. 6 shows a scatter plot of the v2.1 and v1.3 Spring 2019 gridded average $N_v^{strat}$ for all valid pixels with latitude $< 85°$, which shows that the world-wide average increase of $N_v^{strat}$ for this period is about $0.50\,\mu mol\,m^{-2}$. For the other four test periods, the average increase ranges from $0.60$ to $1.55\,\mu mol\,m^{-2}$, all with slopes between $0.99$ and $1.02$. The other three DDS-

2 periods have scatter similar to that seen in Fig. 6, while the Autumn 2020 DDS-3 period has more scatter but still with high

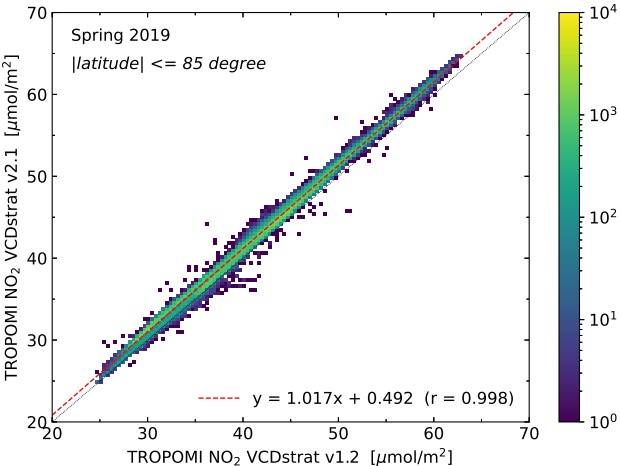

**Figure 6.** Scatter plot of the TROPOMI v1.3 and v2.1 gridded $NO_2$ stratospheric VCD averaged over the Spring 2019 VCD period of all ground pixels with valid retrieval with latitude $\leq 85°$. The linear fit coefficients and correlation coefficient are given in the legend.

correlation ($r = 0.981$). Scatter plots of the gridded GCD (not shown) reveal linear fits with slopes of $1.00$ and offsets ranging from $0.73$ to $1.04 \, \mu\text{mol m}^{-2}$, with little scatter for the four DDS-2 periods and more scatter for the DDS-3 period, all with high correlations ($r > 0.98$). Note that linear fits mentioned in Sect. 3 and 4 are all performed with an Orthogonal Distance Regression (ODR), i.e. taking into account that both data sets have uncertainties, rather than only the data along the y-axis, while in Sect. 5 a different linear regression approach is used.

According to van Geffen et al. (2020) the $NO_2$ SCDs of OMI and TROPOMI agree quite well, with TROPOMI a few percent higher than OMI, as a result of small differences in the DOAS retrieval details, and with OMI showing more scatter than TROPOMI due to its lower spatial distribution. The above described minor changes in the TROPOMI SCD values imply that the same conclusion still holds. Linear fits in scatter plots of world-wide average gridded GCD v2.1 and OMI/QA4ECV data (not shown) for the five test VCD periods have slopes ranging from $0.98$ to $1.03$ and offsets between $1.27$ and $2.57 \, \mu\text{mol m}^{-2}$, with high correlation ($r > 0.94$). A more detailed slant column comparison with OMI, based on regional averages, can be seen in Fig. 14 (Sect. 4.4).

## 4   Updates in the tropospheric VCD step

### 4.1   FRESCO cloud pressure and $NO_2$ cloud fraction

A dedicated version of the FRESCO+ cloud algorithm (Wang at al., 2008), named FRESCO-S (with 'S' for 'Sentinel'), was implemented in the NLL2DP processor to provide a support cloud product, and its cloud pressure data is used for the v1.2-v1.3 $NO_2$ data product. Studies showed that the FRESCO-S cloud pressure is too high for some scenes, in particular for scenes with low cloud fractions and/or a considerable aerosol load (which FRESCO sees as an effective cloud), in which case the cloud

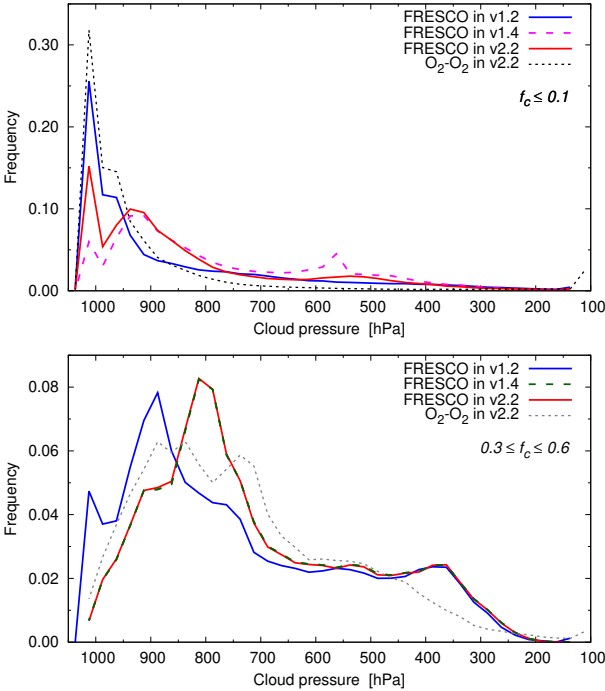

**Figure 7.** Cloud pressure frequency distribution from orbit 03707 on 1 July 2018, considering only ocean and land ground pixels that are free of snow/ice, for small (top panel) and medium (lower panel) cloud fractions, with a bin size of 25 hPa and $f_c$ the cloud fraction in the $NO_2$ fit window. Shown are curves from the FRESCO retrievals in the public v1.2 data (solid blue lines), its implementation in the v1.4 framework with the use of level-1b v1.0 spectra (dashed magenta line), and with the level-1b v2.0, i.e. from DDS-2 (solid red line). Also shown is the cloud pressure frequency from the preliminary $O_2$-$O_2$ retrieval included in the v2.2 processing (dotted black line), which is mentioned in Sect. 6.1.

pressure is close to the surface pressure (cf. Compernolle et al., 2021; Eskes et al., 2021). The consequence of this is that the tropospheric $NO_2$ VCD is too low for these scenes, as shown also in validation comparisons (see the Introduction).

As of $NO_2$ v1.4 the so-called FRESCO-wide approach is used, which provides a more realistic estimate of the cloud pressure for scenes with low cloud fractions: the cloud pressure is lower, i.e. the cloud is higher up, as a result of which the tropospheric

5   AMFs decrease, which in turn leads to higher tropospheric $NO_2$ VCDs. To a large extent, this closes the gap between the TROPOMI and the validation data, though for certain cases a difference between the two datasets remains, as discussed by Eskes et al. (2021); see also van Geffen et al. (2021).

The FRESCO-wide approach is also used for the cloud pressure in the v2.1 (DDS-2) and v2.2 (DDS-3 and its public data release) $NO_2$ data, but with the cloud data retrieved from the improved level-1b v2.0 spectra. Fig. 7 shows the cloud pressure,

10   $c_p$, frequency distribution of a single orbit, considering only ground pixels identified as snow/ice-free land and ocean by the NISE snow/ice flag (cf. Sect. 4.2), in two cloud fraction regimes: small ($f_c \leq 0.1$; top panel) and medium ($0.3 \leq f_c \leq 0.6$; bottom panel), with $f_c$ the cloud fraction in the $NO_2$ fit window. This example shows that the FRESCO-wide implementation

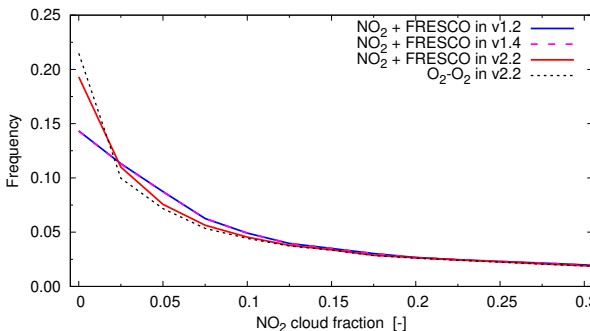

**Figure 8.** Frequency distribution of the cloud fraction in the $NO_2$ fit window from orbit 03707 on 1 July 2018, considering only ocean and land ground pixels that are free of snow/ice, with a bin size of 0.0025 and the first bin centred around 0.0, zooming in on the lower cloud fraction. Shown are curves from the $NO_2$ cloud fraction using the FRESCO cloud pressure in the public v1.2 data (solid blue lines), its implementation in the v1.4 framework with the use of level-1b v1.0 spectra (dashed magenta line), and with the level-1b v2.0, i.e. from DDS-2 (solid red line). Note that the v1.2 and v1.4 lines overlap in the figure: the difference between these two is less than $0.3\,\%$. Also shown is the cloud fraction frequency from the preliminary $O_2$-$O_2$ retrieval included in the v2.2 processing (dotted black line), which is mentioned in Sect. 6.1.

in $NO_2$ v1.4 indeed leads to lower $c_p$. The impact of the use of the level-1b v2.0 in $NO_2$ v2.2 is smaller than the impact of the FRESCO-wide approach for the lowest cloud fractions (top panel) $c_p$, while for the medium (lower panel) and higher (not shown) range cloud fractions no change is visible due to the switch to level-1b v2.0 spectra.

Inspection of the frequency distribution of the $NO_2$ cloud fractions (Fig. 8) reveals an increase in the number of fully cloud-
free pixels in v2.2 at the expense of the number of pixels with small but non-zero cloud fractions due to the use of v2.0 level-1b spectra, while the improvement of the FRESCO cloud pressure in v1.4 has no visible impact on the cloud fraction distributions.

## 4.2   Snow/ice flag

It is important to have information on the presence of snow or ice in a given satellite ground pixel, so that if necessary the climatological surface albedo can be adjusted or the AMF calculation can switch from using the cloud fraction and cloud
pressure to the use of the effective scene pressure and effective scene albedo, because the cloud algorithm has difficulty distinguishing clouds above snow/ice (cf. Eskes et al., 2021; van Geffen et al., 2021) and cloud from sun glint. To this end the v1.2-v1.4 processing uses the daily snow/ice cover database from NISE (Nolin et al., 2005).

The NISE data, however, appear to suffer from a number of problems: it has a rather coarse spatial resolution, for a given day it is based on an average over a few days, and it has problems determining snow/ice content around coastlines. The latter
is in particular problematic at high latitudes where snow/ice coverage may be important. As of v2.1 the snow/ice information is taken from the daily ECMWF meteorological data, which solves the issues with NISE, thus improving the reliability of the $NO_2$ data. Fig. 9 shows an example over Canada of the two snow/ice flag datasets, where the NISE flag numbering is





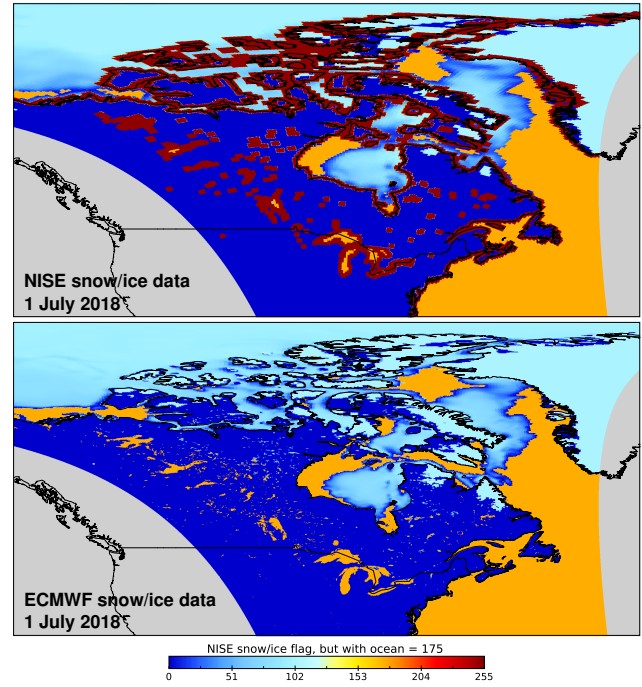

**Figure 9.** Snow/ice flag comparison for the ground pixels of orbits 03707 and 03708 of 1 July 2018 based on NISE (top panel) and ECMWF (bottom panel) snow/ice cover data. The NISE coding for the flags is used, except that "ocean" is coloured with orange (value 175) instead of red for its flag value 255, so as to clearly distinguish it from the flags 252 (mixed pixels at coastlines), 253 (suspect ice value) and 254 (error); the latter three do not occur in the ECMWF data. Other flag meanings: 0 = snow-free land, 1-100 = percentage sea-ice, 101 = permanent ice, 103 = snow. The depicted area is longitude = $[-140° : -40°]$, latitude = $[+35° : +85°]$.

used, except that the ice-free ocean has been given the colour orange (value 175) instead of red for its flag value 255, so as to distinguish it from the problematic NISE flags 252-254.

Another issue solved with the switch to the ECMWF snow/ice data is that the NISE data over shallow water areas that may run dry during low tide can be wrong. Over the western part of the Waddenzee in The Netherlands, for example, NISE gives on 1 Jan. 2019 $3\%$ sea-ice, whereas this area cannot possibly have any sea-ice: the ECMWF data correctly identify pixels as ocean (flag value zero). Because of this corrected identification, the $NO_2$ surface albedo is adjusted from the value of $0.62$ in the climatology to a more realistic $0.04$.

### 4.3 Surface and cloud albedo

The surface albedo in the $NO_2$ fit window, used in e.g. the computation of the cloud fraction and the AMF (van Geffen et al., 2021), is taken from the 5-year version of the OMI Lambertian-equivalent reflectivity (LER) climatology (Kleipool et al., 2008) at $440\,\mathrm{nm}$, which is given on a grid of $0.5° \times 0.5°$.





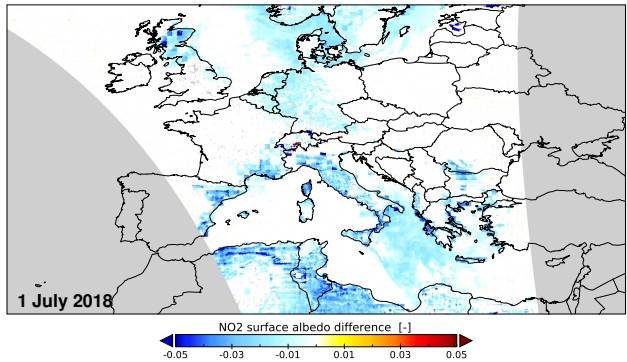

**Figure 10.** Map of the TROPOMI $NO_2$ surface albedo difference "v2.1 minus v1.2" of a part of orbit 03704 of 1 July 2018. See the text for further details. The depicted area is longitude $= [-20^\circ : +40^\circ]$, latitude $= [+30^\circ : +60^\circ]$.

The cloud fraction, $f_c$, is determined in the $NO_2$ fit window at $440\,\mathrm{nm}$ following the same approach the FRESCO cloud retrieval (Wang at al., 2008) uses for the cloud retrieval in the $O_2$ A-band, with an assumed cloud albedo $A_c = 0.8$. On physical grounds $f_c$ lies within the range $[0:1]$. If the actual surface albedo, $A_s$, is lower than expected from the climatology, the cloud retrieval leads to $f_c < 0$. Up to v1.4 this was clipped to zero, whereas as of v2.1 the $A_s$ is adjusted (decreased) to match $f_c = 0$

and ensure radiative closure. Similarly, in case of very bright clouds the cloud retrieval leads to $f_c > 1$, which is no longer clipped but instead the cloud albedo, $A_c$, is adjusted (increased) to reach radiative closure with $f_c = 1$. (For details, see van Geffen et al., 2021, App. C.) This approach of adjusting the surface or cloud albedo to keep the cloud fraction within $[0:1]$ was implemented in the FRESCO cloud retrieval in processor v1.3, leading to more realistic cloud pressures (cf. Eskes and Eichmann, 2021). With the same implementation in use for the $NO_2$ cloud fraction, the treatment is consistent.

Fig. 10 shows as an example a map of the difference "v2.1 minus v1.2" in the $NO_2$ surface albedo for a part of an orbit. A lower surface albedo leads to a smaller AMF and thus to a higher tropospheric $NO_2$ VCD. Fig. 11 shows for the full orbit the relationship between the $NO_2$ tropospheric VCD of v2.1 (vertical axis) and v1.2 (horizontal axis), considering only ground pixels for which the cloud retrieval gives a cloud fraction $f_c < 0.001$ (i.e. effectively zero) and thus for which the surface albedo may have been reduced in v2.1: the tropospheric VCD increases by about $15\,\%$, as the linear fit in Fig. 11 shows. The increase

is larger for higher VCDs: for tropospheric VCDs $< 100\,\mathrm{\mu mol\,m^{-2}}$ the increase is about $10\,\%$ (linear fit: $y = 1.091x - 0.670$). This increase is partly related to the use of level-1b v2.0 (ir)radiance spectra as input in the v2.1 processing; a test processing of the orbit with v2.2 (not shown) reveals that level-1b v2.0 spectra give a tropospheric VCD that is about $5\,\%$ higher than level-1b v1.0 spectra (linear fit over all positive VCDs: $y = 1.053x + 1.233$, correlation coefficient: $r = 0.999$). A similar increase is found when looking at the ground pixels for which the cloud radiance fraction $0.2 < w_c < 0.5$ ($y = 1.059x - 1.135, r = 0.982$).

**4.4   Impact on tropospheric VCD results**

The impact of processor changes on the tropospheric VCD data is dominated by the update of the FRESCO cloud retrieval, mentioned in Sect 4.1, as of v1.4: the other updates in the $NO_2$ processor and the inclusion of the level-1b v2.0 spectra come



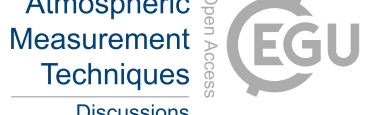

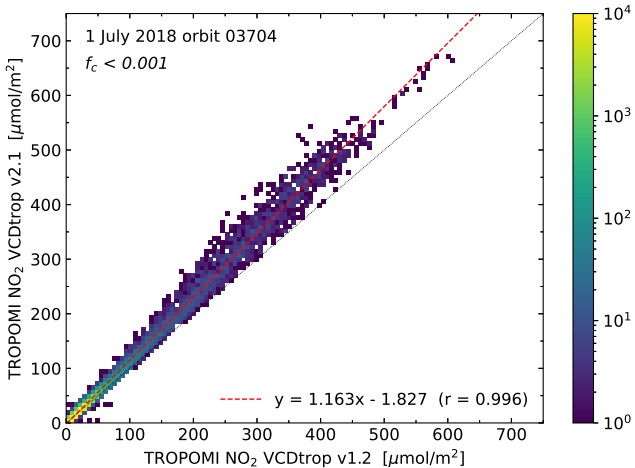

**Figure 11.** Comparison of the TROPOMI $NO_2$ tropospheric VCD of orbit 03704 of 1 July 2018, showing all ground pixels for which both the v1.2 and v2.1 cloud retrieval gives zero cloud fraction, i.e. for which the surface albedo may have been adjusted, and the tropospheric VCD was found to be positive. See the text for further details. The linear fit coefficients and correlation coefficient are given in the legend.

**Table 4.** Results of linear fit and correlation coefficients ($r$) of scatter plots of gridded $NO_2$ tropospheric VCD ($N_v^{\mathrm{trop}}$) averaged over the VCD part of the DDS periods using ground pixels with cloud radiance fraction $w_c < 0.50$ and latitude $\leq 85°$. TROPOMI v1.x or OMI/QA4ECV data is along the $x$ axis and TROPOMI v2.x is along the $y$ axis; the offset, given in µmol m$^{-2}$, is much smaller than typical column values. The first set of TROPOMI data uses grid cells with all $N_v^{\mathrm{trop}}$ values, while the second set and the OMI set use only grid cells with $N_v^{\mathrm{trop}} \leq 100$ µmol m$^{-2}$.

| | | TROPOMI: all $N_v^{\mathrm{trop}}$ | | | TROPOMI: $N_v^{\mathrm{trop}} \leq 100$ | | | OMI: $N_v^{\mathrm{trop}} \leq 100$ | | |
|---|---|---|---|---|---|---|---|---|---|---|
| DDS | Season | slope | offset | $r$ | slope | offset | $r$ | slope | offset | $r$ |
| 2 | Summer 2018 | 1.021 | 0.180 | 0.991 | 1.015 | 0.220 | 0.990 | 0.760 | 1.061 | 0.724 |
| 2 | Winter 2019 | 1.410 | −2.330 | 0.972 | 1.105 | −0.220 | 0.976 | 0.886 | 1.452 | 0.855 |
| 2 | Spring 2019 | 1.155 | −0.738 | 0.984 | 1.102 | −0.390 | 0.985 | 0.847 | 1.498 | 0.846 |
| 2 | Autumn 2019 | 1.108 | −0.252 | 0.991 | 1.090 | −0.120 | 0.992 | 0.835 | 1.525 | 0.885 |
| 3 | Autumn 2020 | 1.157 | −0.895 | 0.988 | 1.106 | −0.499 | 0.989 | 0.880 | 1.124 | 0.894 |

on top of that. Unfortunately none of the DDS periods covers v1.4 data, which means that a differentiation of the results before and after the FRESCO-wide update is not possible.

The bottom row of Fig. 5 shows example maps of the tropospheric VCD ($N_v^{\mathrm{trop}}$) of v2.1 and the difference between DDS and OFFL data, based on all ground pixels with valid retrieval, i.e. including cloudy pixels. For a good comparison of $N_v^{\mathrm{trop}}$

5  data versions it is, however, better to consider only (nearly) cloud-free ground pixels with $w_c < 0.50$ (qa_value $> 0.75$),



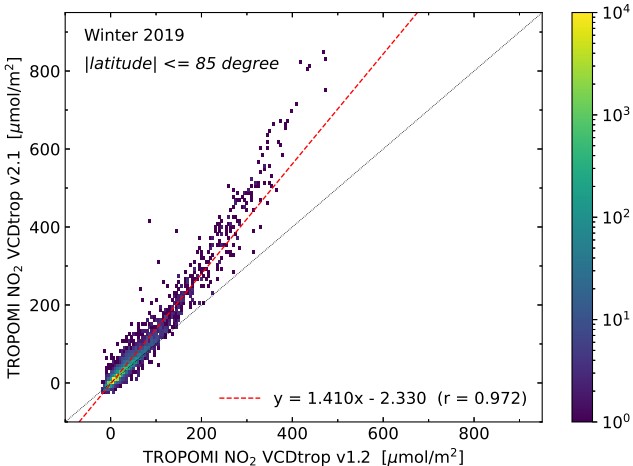

**Figure 12.** Scatter plot of the TROPOMI v1.2 and v2.1 gridded $NO_2$ tropospheric VCD averaged over the Winter 2019 VCD period of ground pixels with cloud radiance fraction $w_c < 0.50$ and latitude $\leq 85°$. The linear fit coefficients and correlation coefficient are given in the legend.

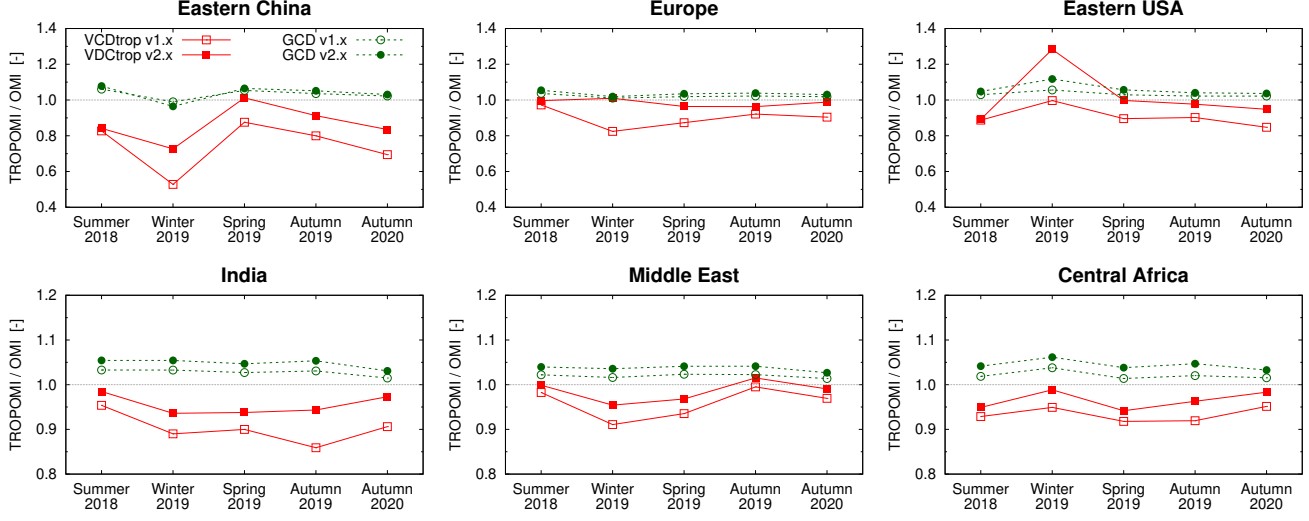

**Figure 14.** Regional averages of the gridded $NO_2$ tropospheric VCD (solid red lines) and GCD (dashed green lines) averaged over the VCD part for the five DDS periods of TROPOMI v1.x (filled symbols) and v2.1 (open symbols) divided by the respective averages of OMI/QA4ECV data. Note the difference in y-axis range of the upper and lower panels. The regions are defined in Table A1.

which amounts to cloud fractions of about 0.2 and less. With this filtering and only 7 or 10 days of data for the average gridded data, the results are somewhat more noisy than those presented in Sect. 3.3.



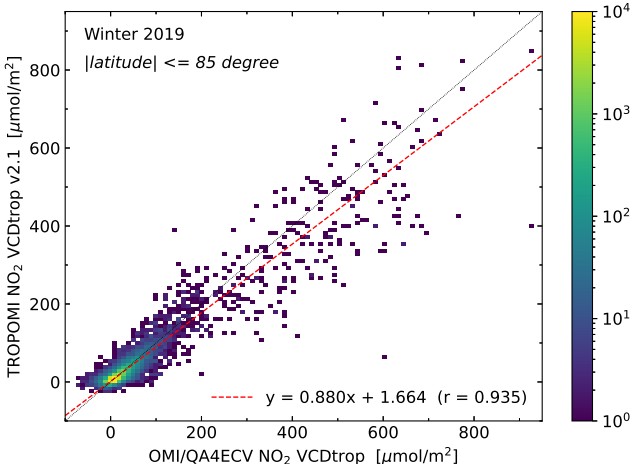

**Figure 13.** As Fig. 12 but for TROPOMI v2.1 data version OMI/QA4ECV data.

Fig. 12 shows a scatter plot of the v2.1 and v1.3 Winter 2019 gridded average $N_\mathrm{v}^\mathrm{trop}$ for all ground pixels with $w_c < 0.50$ and latitude $\leq 85°$. The period is chosen as example because it clearly shows that a linear fit through all the data is dominated by high tropospheric columns: limiting the linear fit to $N_\mathrm{v}^\mathrm{trop} \leq 100\,\mathrm{\mu mol\,m^{-2}}$ gives a slope of $1.105$ rather than $1.410$. Table 4 lists the results for the linear fits of the five DDS periods, as in Fig. 12, for all $N_\mathrm{v}^\mathrm{trop}$ and for $N_\mathrm{v}^\mathrm{trop} \leq 100\,\mathrm{\mu mol\,m^{-2}}$.

From Fig. 12 and Table 4 it is clear that the average $N_\mathrm{v}^\mathrm{trop}$ increases with the improvements in the algorithm. With this increase, the TROPOMI data lies closer to the OMI/QA4ECV tropospheric VCD, as shown in Fig. 13 and the last three columns of Table 4. To further investigate the changes in the TROPOMI data, Fig. 14 shows comparisons of averages over selected regions (defined in Table A1) of the gridded tropospheric VCD (red solid lines) and the GCD (green dashed lines), where the TROPOMI averages are divided by the OMI/QA4ECV averages. Clearly, TROPOMI v2.x gives higher tropospheric

VCDs than v1.x, in particular for the winter periods in polluted areas (upper panels in Fig. 14). In most cases the TROPOMI v2.x tropospheric VCD lies closer to OMI than TROPOMI v1.2.

## 5   Ground-based validation

To assess the impact of the processor changes on the $NO_2$ VCD data through ground-based validation, both the operational OFFL and the updated DDS data of the five DDS periods for which VCD data is available (cf. Table 1) are compared for three

sets of ground-based measurements provided by monitoring networks:

- $NO_2$ stratospheric column data measured by Zenith-Scattered-Light Differential Optical Absorption Spectroscopy (ZSL-DOAS) instruments from the Network for the Detection of Atmospheric Composition Change (NDACC) (Solomon et al., 1987; Pommereau and Goutail, 1988; Kreher et al., 2020),


– NO$_2$ tropospheric column data from Multi-Axis DOAS (MAX-DOAS) instruments (Hönninger et al., 2004; Hendrick et al., 2014; Kanaya et al., 2014; Kreher et al., 2020; Pinardi et al., 2020),

– NO$_2$ total column data from Pandora direct-sun instruments (Herman et al., 2009, 2019) from the Pandonia Global Network (PGN).

The validation approach is described in Verhoelst et al. (2021). For practical reasons, only the NDACC ZSL-DOAS measurements acquired at sunset were used here for the stratospheric comparisons, after a model-based photo-chemical adjustment of the ground-based twilight column to the satellite overpass time. To ensure robustness of the validation results despite the small size of the DDS data periods, only sites offering at least 5 co-located data pairs were retained. No harmonisation using averaging kernels nor a priori profiles was performed. More details on ground-based data sets, including station details, and

the comparison methodology can be found in Verhoelst et al. (2021); Compernolle et al. (2020); Pinardi et al. (2020); Kumar et al. (2020).

Ground-based validation results are presented here in the commonly used unit Pmolec cm$^{-2}$, where $1.0 \times 10^{15}$ molec cm$^{-2}$ is equal to $1.660539 \times 10^{-5}$ mol m$^{-2}$ in SI units. Contrary to the use of ODR for linear fits in Sect. 3 and 4, here two cases of ordinary least squares fits are used: $y$ vs $x$ and $x$ vs $y$, thus considering two limiting cases of attributing all error variances to $y$

and $x$, respectively.

## 5.1    Stratospheric column

The ground-based validation of the stratospheric NO$_2$ column data reveals an improvement in the bias, from a median difference over all co-located pairs of $-0.2$ Pmolec cm$^{-2}$ (identical to the bias reported in Verhoelst et al., 2021 and amounting to about $-6\%$) for the operational OFFL data, to $-0.1$ Pmolec cm$^{-2}$ ($-3\%$) for the updated DDS data, which is in line with the

slight increase in TROPOMI stratospheric column mentioned in Sect. 3.3. Typically, stratospheric columns show a seasonal variation between 2 and 3 Pmolec cm$^{-2}$ for nearly equatorial sites and between 1 and 6 Pmolec cm$^{-2}$ for sites at very high latitudes. The updated processing does not change significantly the correlation (Pearson-R) and the dispersion (half of the 68 interpercentile) of the difference between ground-based and S5P stratospheric column data, and only slightly the results of a linear regression (see Fig. 15a), as expected from the reduced bias.

Investigating results at individual ground stations (Fig. 15b) shows improvements in bias (reduction of the absolute value of the median difference) at 6 out of 9 stations, almost no change for one, and increases for the last two stations. The large bias at the Ny-Ålesund station, at about 79° N, is under investigation; other high-latitude stations, for which there unfortunately were no co-locations in the DDS periods, do not show such a large bias.

## 5.2    Tropospheric column

The comparison of TROPOMI to MAX-DOAS tropospheric NO$_2$ column data reveals an improvement in both the bias and the dispersion. The former improves from a median difference over all co-located pairs of $-1.4$ Pmolec cm$^{-2}$ (or about $-32\%$ and similar to the bias reported in Verhoelst et al., 2021) for the OFFL to $-0.9$ Pmolec cm$^{-2}$ ($-23\%$) for the DDS data.



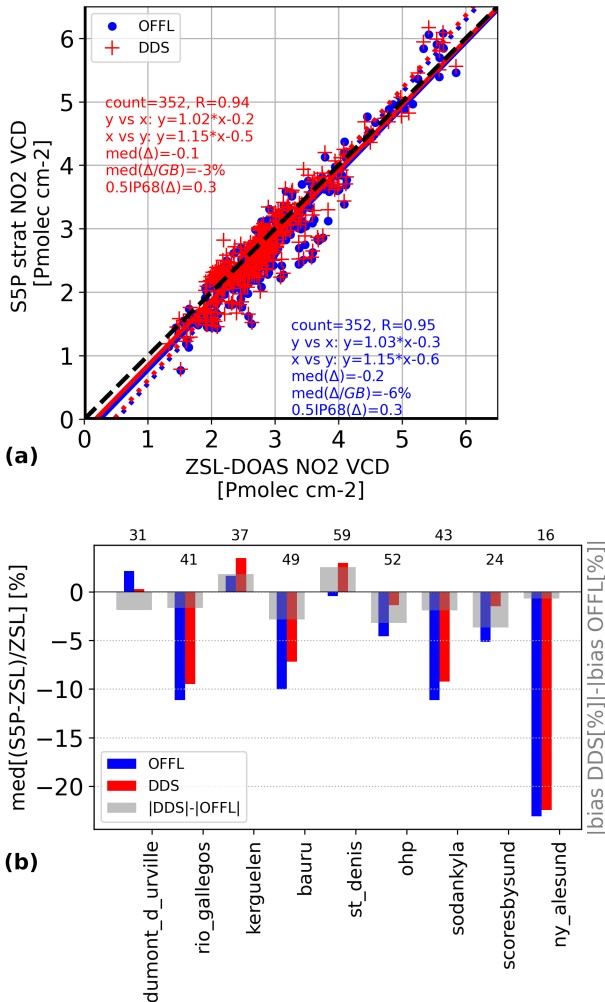

**Figure 15. (a)** Correlation between S5P/TROPOMI and ZSL-DOAS stratospheric $NO_2$ column number densities, after photochemical adjustment of the latter to the satellite overpass time. The operational OFFL data are presented in blue, the reprocessed DDS in red. Ordinary linear regression results are indicated for both $y$ vs $x$ (solid lines) and $x$ vs $y$ (dotted lines). **(b)** Median relative difference (bias) between S5P/TROPOMI and ZSL-DOAS stratospheric $NO_2$ column number densities, per station, for both the DDS data (red) and the corresponding OFFL data (blue). The change of the median relative difference is indicated in gray. The number of co-located pairs at each station is provided along the top axis. Stations are ordered per increasing latitude.

The dispersion of the difference improves from $3.3\,\mathrm{Pmolec\,cm^{-2}}$ to $2.4\,\mathrm{Pmolec\,cm^{-2}}$. Fig. 16a demonstrates that also the linear regression improves somewhat, with a slight increase of the slope, as expected from the improvement in the derived (multiplicative) bias.



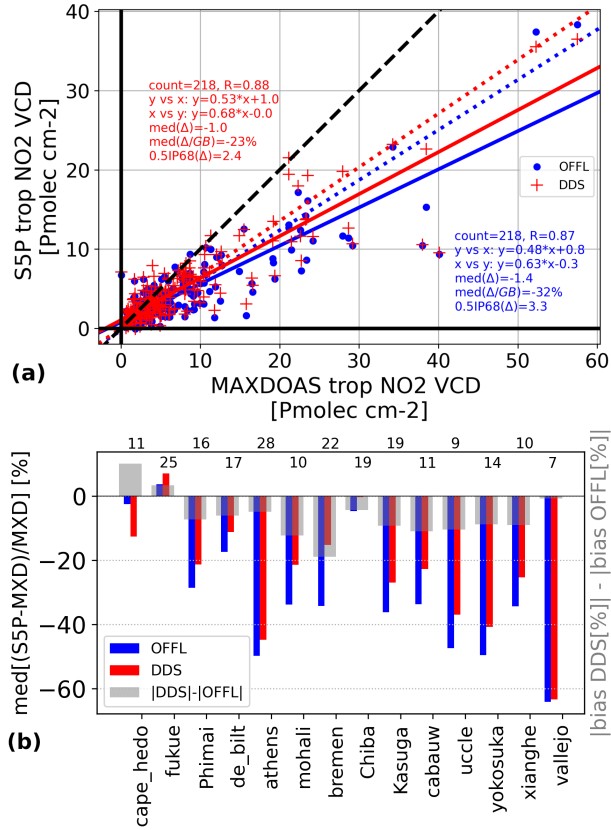

**Figure 16.** Similar to Fig. 15 but for S5P/TROPOMI versus MAX-DOAS tropospheric NO$_2$ column measurements, which require no photochemical adjustment. Sites in panel (b) are ordered along increasing MAX-DOAS tropospheric column.

Looking at the change in bias at each individual station (Fig. 16b), the DDS data shows lower biases than the OFFL data at all but two of the 16 stations. However, results at these two outlying sites cannot be considered as meaningful: they represent relatively clean background conditions with small tropospheric column values, with already very small biases in the OFFL data. It is interesting to note that the improvement does not scale with the tropospheric column value: the most polluted site

5   does not benefit from a larger improvement.

## 5.3 Total column

Similar to the tropospheric column validation, the comparison of TROPOMI to PGN total column NO$_2$ data reveals an improvement in both the bias and the dispersion. The former improves from a median difference over all co-located pairs of $-0.8$ Pmolec cm$^{-2}$ (or about $-12\%$) for the OFFL data to $-0.3$ Pmolec cm$^{-2}$ ($-5\%$) for the DDS data. The difference dis-

10   persion improves slightly from $2.5$ Pmolec cm$^{-2}$ to $2.3$ Pmolec cm$^{-2}$. Fig. 17a shows that also the linear regression improves somewhat, with a clear increase of the slope, as expected from the improvement in the derived (multiplicative) bias.





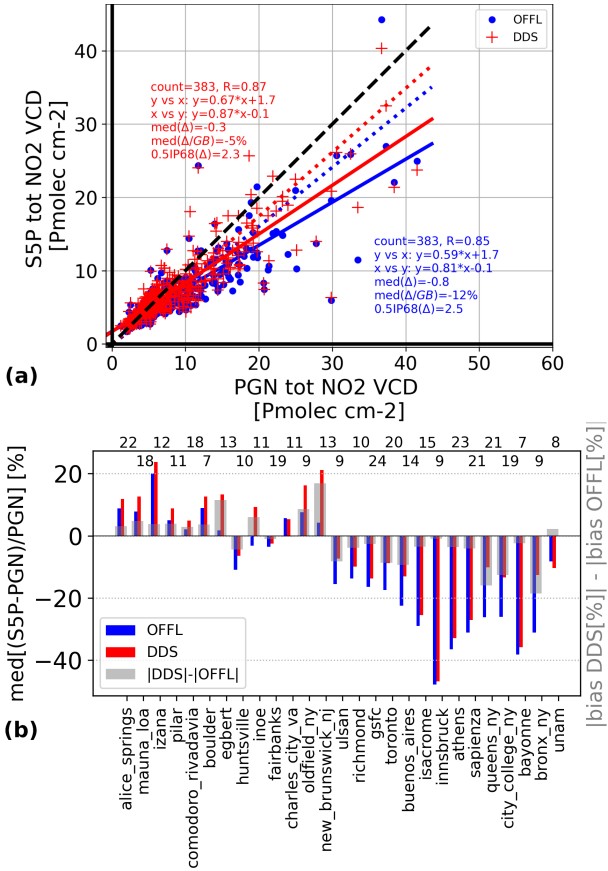

(a)

(b)

**Figure 17.** Similar to Fig. 15 but for S5P/TROPOMI versus PGN total $NO_2$ column measurements, which require no photochemical adjustment. Sites in panel (b) are ordered along increasing PGN total column.

Looking at the bias per station (Fig. 17b), the situation is more complex to describe than for the tropospheric column. At relatively clean sites with small tropospheric column values, for which the OFFL data already presented slight positive biases with regard to the PGN measurements, the increased columns in the DDS data lead to even larger positive biases. The increased DDS total columns improve the bias only of those sites for which the OFFL data underestimated the PGN columns. This is

5 also broadly true for the right-hand half of the graph, i.e. for sites with larger total columns due to a significant tropospheric contribution, in line with the findings for the MAX-DOAS comparisons (where the network is heavily biased towards sites with significant tropospheric columns).





## 5.4 Validation summary and discussion

In summary, ground-based validation of the updated DDS $NO_2$ vertical column data, in comparison to the validation of the corresponding operational OFFL data, confirms the improvement (reduction) in the bias of the stratospheric column expected from the increase in the stratospheric column observed in the DDS data.

For the tropospheric and total $NO_2$ columns, the dispersion is lower with the DDS data, but whether the bias improves depends on the range of tropospheric column values: at sites with large tropospheric columns, affected by strong negative biases in the OFFL data, the increased tropospheric (and total) columns imply a clear improvement. At clean background sites, however, the increased columns of the DDS data actually worsen the already positive bias of the OFFL data, a finding which is somewhat at odds with the ZSL-DOAS comparisons for the stratospheric columns, where an originally negative bias

is reduced. This apparent inconsistency between direct-sun and zenith-sky measurements was already observed in Verhoelst et al. (2021) and work is ongoing to elucidate and address this, including a reprocessing of the PGN data (upcoming v1.8) with more appropriate absorption cross-sections for clean sites where the total column resides mostly in the stratosphere.

Note that the two most polluted measurement sites, which show in the tropospheric (Fig. 16b: Vallejo) and total (Fig. 17b: Unam) column bias a behaviour very different from the other polluted sites, are both located on the Mexican plateau, a situation

very different from the other measurement sites.

## 6 Planned improvements beyond data version 2.2

### 6.1 $O_2$-$O_2$ cloud data retrieval

Processor version v2.2, used for DDS-3 and operational since 1 July 2021, includes in the $NO_2$ processing chain an implementation of the $O_2$-$O_2$ cloud data retrieval algorithm used for OMI and described by Veefkind et al. (2016), which is based on a

DOAS SCD retrieval of the absorption feature of the $O_2$-$O_2$ collision-complex in the wavelength window $[460 : 490]$ nm. The results of this cloud retrieval are included in the standard $NO_2$ data file, but are not further used yet: they are still under evaluation for fine-tuning the algorithm settings. As an example, the $O_2$-$O_2$ cloud pressure and cloud fraction frequency distribution is included in Figs. 7 and 8.

The $NO_2$ cloud (radiance) fraction is currently derived from the FRESCO cloud pressure, as mentioned in Sect. 4.1. Using

the $O_2$-$O_2$ cloud pressure instead would mean that the cloud pressure is determined a) from almost the same wavelengths as $NO_2$ and b) from measurements by the same detector, thus eliminating the small spatial mismatch between ground pixels of $NO_2$ (band 4) and FRESCO (band 6). In addition it seems that for certain atmospheric circumstances the $O_2$-$O_2$ cloud pressure may be more realistic than the FRESCO cloud pressure. Evaluation of the $O_2$-$O_2$ cloud data product quality is ongoing, which may lead to selections rules within the $NO_2$ processor to choose between the two cloud pressures. It is as yet uncertain whether

this will be included in data processor version 2.4.



## 6.2 Bug fixes in data version v2.3

Fixes have been included in data processor version v2.3, operational as of 14 Nov. 2021, of minor bugs related to the output of some detailed data not used by most data users (notably wavelength calibration parameters and $NO_2$ DOAS polynomial coefficients) that were accidentally introduced in v2.2 with the inclusion of the $O_2$-$O_2$ cloud retrieval and which do not affect
the v2.2 SCD and VCD values or quality.

## 6.3 Level-1b radiance degradation correction

The improvements in the level-1b v2 spectra (cf. Sect. 2.1.3; Ludewig et al., 2020) include a correction for the degradation of the irradiance but not for the radiance, because at the time of delivery of the initial level-1b v2.0 calibration key data (CKD) the accumulated degradation in radiance was still to small to reliably determine a degradation correction for it. With a stronger
effect and more radiance data data available, it has become possible to determine a degradation correction and updated CKD have been determined.

    With this update a new test data set, DDS-4, is made in Autumn 2021. If evaluation of TROPOMI data products in DDS-4 is favourable, the radiance degradation correction will be included in the operational processor, which for $NO_2$ will be v2.4, due for release in the first half of 2022 and due to be used for a full mission reprocessing later in 2022. Little effect is expected
on the $NO_2$ SCD values, but the SCD error may improve due to the degradation correction.

## 6.4 TROPOMI surface albedo data

As mentioned in Sect. 4.3 the surface albedo in the $NO_2$ fit window is taken from the 5-year version of the OMI Lambertian-equivalent reflectivity (LER) climatology (Kleipool et al., 2008), which is given on a grid of $0.5° \times 0.5°$ and measured at almost the same overpass time as TROPOMI is measuring. The OMI LER, however, does not contain NIR wavelengths and for the
FRESCO cloud retrieval the GOME-2 LER (Tilstra et al., 2017) is used, which is given on a grid of $0.25° \times 0.25°$ and measured at mid-morning rather than early afternoon. These climatologies are not optimal for TROPOMI, in particular in view of the spatial resolution. Furthermore, the LER approach assumes isotropic reflection of light, while in reality there is a viewing angle dependency in the reflected light (see e.g. Lorente et al., 2018).

    For this reason a dedicated TROPOMI surface albedo climatology is under development, based on TROPOMI measurements
and containing both a traditional LER as well as a directionally dependent LER (DLER), similar to the one developed recently from GOME-2 measurements by Tilstra et al. (2021). This TROPOMI climatology will be available at a grid of $0.125° \times 0.125°$. Initially it will be based on level-1b v1.0 spectra and as such it is expected to be available for use in both the FRESCO and $NO_2$ v2.4 operation processing and planned mission reprocessing. At a later stage, after the mission reprocessing, an update of the TROPOMI climatology will be made using level-1b v2.0 spectra. Whether and if so when that updated DLER will be
implemented in the FRESCO and $NO_2$ processing is as yet undecided.





# 7  Concluding remarks

The TROPOMI $NO_2$ data product is widely used for monitoring air pollution levels world-wide, benefiting from TROPOMI's high spatial sampling and excellent signal-to-noise ratio. Since the first data release mid 2018 several improvements have been made, with a major update to version 1.4 at the end of November 2020 (van Geffen et al., 2020, 2021; Eskes et al., 2021; Eskes

and Eichmann, 2021). This paper documents the improvements leading to version 2.2 of the TROPOMI $NO_2$ data product, operational as of 1 July 2021. These improvements and their impact on the $NO_2$ SCD and VCD data, studied by comparing so-called Diagnostic Data Set (DDS) test data with operational offline (OFFL) v1.x data, can be summarised as follows.

- Small corrections in the wavelength assignment of the reflectance used in the DOAS slant column fit reduce the SCD error of ground pixels along some detector rows, without affecting other rows or the SCD values significantly.

- The introduction of an outlier removal improves the SCD retrieval quality for ground pixels suffering from charged particles hitting the detector (notably over the SAA) and those suffering from saturation and blooming effects (notably over bright clouds), without affecting other ground pixels.

- The use of improved level-1b v2.0 (ir)radiance spectra, with among others better handling of blooming and transients effects, improved the (ir)radiance calibration, and improved irradiance degradation correction, in combination with the

above two improvements, leads to: a) a reduction of the SCD error by about $2\%$, b) a reduction of the RMS error of the DOAS fit by about $7\%$, and c) an increase of the SCD values of about $3\%$.

- The increase of the SCD values is fairly homogeneous and leads to an estimated increase of the stratospheric VCD by $2-4\%$ or $0.6-1.5\,\mu\mathrm{mol\,m^{-2}}$.

- The use of the improved level-1b v2.0 leads a) to a somewhat lower cloud pressure for ground pixels with small clouds

fractions, which in turn leads to tropospheric VCDs for those ground pixels to be higher by some $5\%$, and b) to a small increase of the number fully cloud-free ground pixels.

- Switching the source of the snow/ice flag from NISE to ECMWF improves the quality of the VCD data because of the higher spatial resolution of the ECMWF flag and its better handling of coastlines and shallow water cases.

- The climatological surface albedo reduction for cloud-free ground pixels with reflectances lower than expected, in com-

bination with the use of improved level-1b v2.0 spectra, leads to tropospheric VCDs to be higher by $10-15\%$ for cloud-free pixels.

The combined effect of all improvements on the vertical column data necessarily includes the impact of an update of the FRESCO cloud retrieval as of v1.4 (Eskes et al., 2021) since there is no DDS that covers v1.4 data. On average the v2.x DDS data have tropospheric $NO_2$ columns that are 10 to $40\%$ larger than the v1.x OFFL data, depending on the level of pollution.

This increase has brought these VCDs closer to OMI observations, while the underlying SCDs differ by only a few percent.





Ground-based validation of the updated DDS $NO_2$ vertical column data, in comparison to the validation of the corresponding operational OFFL data, shows on average an improvement of the negative bias of the stratospheric (from $-6\%$ for OFFL to $-3\%$ for DDS), tropospheric (from $-32\%$ to $-23\%$) and total (from $-12\%$ to $-5\%$) columns. For individual measurement stations, however, the picture is more complicated, in particular for the tropospheric and total columns. For most polluted

sites the negative bias improves, but improvement is not proportional to the pollution level. And at clean background sites the positive bias seems to get worse, which in turn seems inconsistent with the improved bias in the stratospheric column. Work is ongoing to try to clarify these differences.

Part of the negative bias observed when comparing with ground-based observations is probably due to the relatively coarse $(1° \times 1°)$ resolution of the a-priori profiles used in the retrieval. Douros et al. (2021) show that the use of profiles from the

CAMS $0.1° \times 0.1°$ air-quality analyses leads to substantial increases of the retrieved tropospheric columns over emission hotspots of order $20\%$, depending on the location.

Processor version 2.3, operational since 14 Nov. 2021, contains only fixes of minor bugs not affecting the SCD or VCD data. Version 2.4, due for release in the first half of 2022 and to be used for a full mission reprocessing later in 2022, may contain a few further improvements, depending on upcoming analyses: a) level-1b spectra with a radiance degradation correction, b) use

of a dedicated TROPOMI surface albedo climatology in both the cloud data and $NO_2$ retrieval that accounts for viewing angle dependencies, and c) criteria in the determination of the $NO_2$ cloud (radiance) fraction between use of the cloud pressure from the FRESCO or from the $O_2$-$O_2$ cloud data product.

## Appendix A: Region definitions

Table A1 gives the longitudinal and latitudinal extent of the regions in Fig. 14.

**Table A1.** Definition of the regions in Fig. 14.

| Region | longitude range | latitude range |
|---|---|---|
| Eastern Chins | $+110.0° : +124.0°$ | $+21.0° : +43.0°$ |
| Europe | $-10.0° : +25.0°$ | $+35.0° : +60.0°$ |
| Eastern USA | $-89.0° : -69.0°$ | $+32.0° : +48.0°$ |
| India | $+69.0° : +89.0°$ | $+8.0° : +34.0°$ |
| Middle East | $+30.0° : +60.0°$ | $+15.0° : +40.0°$ |
| Central Africa | $-17.0° : +37.0°$ | $+4.0° : +18.0°$ |

*Author contributions.* JvG conducted the research described in this paper and is responsible for the text. HE is responsible for the AMF and VCD steps and the final data product. SC, GP, TV, and JCL carried out the global validation analysis. MS and MtL implemented and tested



the retrieval code in the TROPOMI processor. AL is leader of the TROPOMI level-1b team. KFB is involved in the final NO$_2$ data product. JPV is involved in retrieval issues and is the PI of TROPOMI.

*Competing interests.* The authors declare that they have no conflict of interests.

*Data availability.* Standard TROPOMI NO$_2$ data (v1.2-1.4) are available via ESA's public data hub (https://s5phub.copernicus.eu/). Diagnostic data sets DDS-2 (v2.1) and DDS-3 (v2.2) TROPOMI NO$_2$ data are available for registered users via ESA's expert data hub (https://s5pexp.copernicus.eu/). Orbits processed with non-standard configuration are available from the lead author upon request. OMI/QA4ECV NO$_2$ (v1.1) data are available via the QA4ECV portal (http://www.qa4ecv.eu/). The ZSL-DOAS and Pandora data are obtained as part of the Network for the Detection of Atmospheric Composition Change (NDACC, https://ndacc.org/) and the Pandonia Global Network (PGN, https://www.pandonia-global-network.org/), respectively, and are publicly available, while part of the MAX-DOAS data is also available via NDACC.

*Acknowledgements.* Part of the reported work was carried out in the framework of the Copernicus Sentinel-5 Precursor Mission Performance Centre (S5P MPC), contracted by the European Space Agency (ESA/ESRIN, Contract No. 4000117151/16/I-LG) and supported by the Belgian Federal Science Policy Office (BELSPO), the Royal Belgian Institute for Space Aeronomy (BIRA-IASB), the Netherlands Space Office (NSO), and the German Aerospace Centre (DLR). The authors are grateful to ESA/ESRIN for supporting the ESA Validation Data Centre (EVDC) established at NILU, and for running the Fiducial Reference Measurements (FRM) programme and in particular the FRM4DOAS and Pandonia projects.

Part of this work was carried out in the framework of the S5P Validation Team (S5PVT) AO projects NIDFORVAL (ID #28607, PI G. Pinardi, BIRA-IASB) and CESAR (ID #28596, PI A. Apituley, KNMI). S. Compernolle, G. Pinardi and T. Verhoelst at BIRA-IASB acknowledge national funding from BELSPO and ESA through the ProDEx projects TROVA-E2 (PEA 4000116692). The authors express special thanks to A.M. Fjæraa, J. Granville, S. Niemeijer, and O. Rasson for post-processing of the network and satellite data and for their dedication to the S5P/TROPOMI operational validation.

A. Pazmiño, A. Bazureau, F. Goutail, J.-P. Pommereau are acknowledged for the fast delivery of ZSL-DOAS SAOZ data with the LATMOS Real-Time processing facility, and the PIs and staff at stations from LATMOS/CNRS and NILU for operating SAOZ instruments. The SAOZ network received funding from the French Institut National des Sciences de l'Univers (INSU) of the Centre National de la Recherche Scientifique (CNRS), Centre National d'Etudes Spatiales (CNES) and Institut polaire fraçais Paul Emile Victor (IPEV).

The MAX-DOAS data used in this publication were obtained from J.P. Burrows, M. Grutter, H. Irie, Y. Kanaya, A. Piters, M. Van Roozendael, V. Sinha, T. Wagner. Part of the MAX-DOAS data used here is available at the Network for the Detection of Atmospheric Composition Change (NDACC). Fast delivery of MAX-DOAS data tailored to the S5P validation was organized through the S5PVT AO project NIDFORVAL. IUP-Bremen ground-based measurements are funded by DLR-Bonn received through project 50EE1709A. We acknowledge the IISER Mohali atmospheric chemistry facility for supporting the MAX-DOAS measurements at Mohali, India. KNMI ground-based measurements



in De Bilt and Cabauw are partly supported by the project "Ruisdael Observatory", Dutch Research Council (NWO) contract 184.034.015, by the Netherlands Space Office (NSO) for Sentinel-5p/TROPOMI validation, and by ESA via the EU CAMS-27 project.

We thank the PIs, support staff and funding for establishing and maintaining Pandora instruments at the 27 sites of the PGN used in this investigation, from institutes AEMET, CCNY, CITEDEF, ECCC, EPA, ESA, GA, INOE, LUFTBLICK, NASA.GSFC, NOAA.ESRL,
5 PMOD.WRC, UAF, UAH, UC.BERKELEY, UNAM, UNIST and VCU. The PGN is a bilateral project supported with funding from NASA and ESA.

The authors would further like to thank the following people: Erwin Loots and Emiel van der Plas on level-1b issues, Gijsbert Tilstra for on surface albedo issues, and Piet Stammes on general retrieval issues.

Sentinel-5 Precursor is a European Space Agency (ESA) mission on behalf of the European Commission (EC). The TROPOMI payload
10 is a joint development by ESA and the Netherlands Space Office (NSO). The Sentinel-5 Precursor ground-segment development has been funded by ESA and with national contributions from The Netherlands, Germany, and Belgium. This work contains modified Copernicus Sentinel-5P TROPOMI data (2018-2021), processed in the operational framework or locally at KNMI, with post-processing for validation purposes performed by BIRA-IASB.



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
