# Peer review of "Sentinel-5P TROPOMI NO2 retrieval: impact of version v2.2 improvements and comparisons with OMI and ground-based data"

_Atmospheric Measurement Techniques, 2021_

## Author Comment (AC1)

**Reply to Comment on amt-2021-329 by Anonymous Referee #1**

Referee comment on "Sentinel-5P TROPOMI NO2 retrieval: impact of version v2.2 improvements and comparisons with OMI and ground-based data" by Jos van Geffen et al., Atmos. Meas. Tech. Discuss., https://doi.org/10.5194/amt-2021-329-RC1, 2021

 $\implies$  The referee report is copied below; the reply is preceded by an arrow, like this text.

In this manuscript, the authors report on the latest versions of the operational TROPOMI NO2 retrieval, the changes relative to earlier versions and the effect they have on NO2 slant columns as well as vertical tropospheric and stratospheric columns for a so-called diagnostic data set (DDS). The v2.2 columns are then validated by comparison to groundbased zenith-sky, MAX-DOAS and Pandora observations and systematic improvements are found compared to the offline product in most cases.

The TROPOMI NO2 product is widely used in the scientific community and detailed description of algorithm changes and their impacts on the product are of interest to many users. The manuscript is overall well written, clearly structured and reports on relevant results and therefore should be published.

 $\implies$  We thank the referee for these kind words.

I am however surprised by the general lack of discussion of the results throughout the text and hope that the authors can improve on this in the revised version, taking into account the suggestions made below.

I also think that it is very unfortunate that separation of effects was apparently not possible for the cloud effects, which probably are the most relevant and would deserve a more detailed discussion. It is hard to believe that such an important change to the product was introduced without having an extended period of data processed with both FRESCO versions for comparison. Maybe this will be discussed in the manuscript by Eskes et al. (in preparation), but for the manuscript as it stands, this is a clear shortcoming which should be fixed.

 $\implies$  It is indeed unfortunate that we cannot get a better understanding of the impact of the change in the FRESCO algorithm on the NO2 results: as the reviewer correctly writes there is no period for which both the original FRESCO of v1.2/v1.3 and the FRESCO-wide adaptation of v1.4 is available, without any other changes in the NO2 processing. The analyses given in this paper and also in Riess et al. (2022), comparing v1.2/v1.3 data with FRESCO-S and DDS data with FRESCO-wide, however, serves as a very good indicator because the impact on the NO2 data is dominated by the changes in the cloud pressure as a result of the update to FRESCO-wide, and this cloud pressure change can be and is investigated directly.

**Major comments**

**Discussion of new FRESCO version:**

• As mentioned above, a scientific sound comparison separates independent effects and I therefore expect to see a comparison of NO2 columns for at least a few days where only the FRESCO version differs.

⇒ As mentioned above this is not possible because we do not have NO2 data for both FRESCO versions without other changes in the NO2 processing. For Figs. 7 and 8 one orbit was processed locally with different versions, but that is a somewhat artificial approach as it is not only the FRESCO-version itself that changes, but it gives a good indication of the impart. Doing similar calculation locally for multiple days is beyond our possibilities. The study by Riess et al. (2022) gives further analysis of the cloud pressure changes. That study reports on 50 hPa lower cloud pressures, lower AMFs, and up to  $2 \times 10^{15}$  molec.cm-2 higher NO2 columns, resulting from the improved FRESCO algorithm (Fig. 6, 7 and Table 3).

• The discussion of Figure 7 is very superficial – although it is clear from the picture that high cloud pressures have again become more frequent from v1.4 to 2.2 this is not discussed.

 $\implies$  There is a small change visible for the lowest cloud fractions in the upper panel of Fig. 7 between the v1.4 and v2.2 curves, due to the impact of the use of level-1b v2.0 spectra in the latter, as mentioned in the text. For the other cloud fraction ranges there is no such difference: in the lower panel of Fig. 7 there is no difference visible between the v1.4 and v2.2 curves. Evidently, the use of level-1b v2.0 spectra leads to somewhat more low altitude clouds in case of low cloud fractions. The text in Sect. 4.1 has been adapted to make this clearer:

... The use of the level-1b v2.0 in NO2 v2.2 has a smaller impact than the implementation of the FRESCO-wide approach for the lowest cloud fractions (top panel) and appears to lead to somewhat higher cloud pressures in that range. For the medium (lower panel) and high (not shown) range cloud fractions no change is visible due to the switch to level-1b v2.0 spectra.

• Also shown in the figure are results for O2-O2 which agree much better with FRESCO v1.2 than with FRESCO v2.2, but again this is not even mentioned. If you do not trust the O2-O2 results, then remove them. If you show them, then please discuss them.

 $\implies$  Given that the O2-O2 results are not really relevant for the present paper and apparently lead to confusion, it has been decided to remove these results from Figs. 7 and 8. Evaluation of the O2-O2 results is still on-going by others, and it is not yet known when these results may be used for (part of) the NO2 retrieval. In Sect. 6.1 the following has been added:

 $\dots$  Riess et al. (2022) included  $O_2$ - $O_2$  cloud data from OMI in their comparison of FRESCO-S and FRESCO-wide cloud pressures with those of VIIRS (the Visible/Infrared Imager/Radiometer Suite on board of the SUOMI National Polar-orbiting Partnership (SNPP) satellite):  $O_2$ - $O_2$  cloud pressures are systematically higher at low cloud fractions.  $\dots$

**Discussion of surface albedo adjustment:**

I think that this is a very good idea and it is nice that it has been implemented. However, there is hardly any discussion provided in the text although many questions come to mind:

• How will radiance calibration issues and the known low bias in TROPOMI radiances impact on this correction?

 $\implies$  The surface albedo adjustment corrects for the known issue of non-closure in the radiance budget of the AMF approach. Our adjustment is a fundamental, theoretical improvement in the AMF-calculation. Whether and how radiance degradation influences the adjustment will be investigated when more is known about TROPOMI radiance degradation. It is important here to make a distinction between FRESCO (NIR) and NO2 (VIS). In the NIR the degradation is a lot smaller than in the VIS. So for the cloud retrieval itself the impact is negligible, for FRESCO the general properties of the used LER database dominate (this will be corrected in a future release, version 2.4, where a Tropomi-derived DLER will be used). For the VIS the degradation is significant, and this can impact the AMF calculation or the surface albedo correction. Again please note that the surface albedo database used here probably has a bigger impact than the radiance degradation itself. The upcoming L1B release (version 2.1) will include a radiance degradation correction, which will mitigate the issue.

• What about absorbing aerosols?

 $\implies$  In case of high aerosol load, the FRESCO algorithm will detect that as a cloud with non-zero, positive cloud fraction. In this case there is therefore no adjustment of the surface albedo. This non-zero cloud fraction is used in the AMF calculation and sort of accounts for the aerosol load, which is not corrected for in another way. This is the "standard" way of dealing with aerosol load and a further discussion falls outside the scope of the paper.

• Are the patterns found in the albedo correction stable over time and are they plausible in magnitude and pattern?

 $\implies$  Diagnositics of the albedo correction applied is not available, as that is not written to the output files. A few cases, like the one depicted in Fig. 10, have been looked at, comparing the final albedo with the one given in the climatology. A further detailed analysis might be interesting but falls outside the scope of the present paper.

• What is the reason for the bias? Is it because the processor uses the mode instead of the

minimum in the OMI reflectance data base? Is it because of the low bias in TROPOMI radiances? Is it because of cloud shadows? Or does BRDF play a role?

 $\implies$  The surface albedo database used for FRESCO is based on GOME-2, with a different observation time compared to Tropomi. This alone will lead to systematic differences between the best value for TROPOMI and what we currently have. The different spatial resolution will also have a significant impact. For NO2 the OMI surface albedo is used, so the difference in the observation time is much less of a concern. The spatial resolution certainly has an impact, both in the statistical cloud removal in the source data as used for the OMI albedo database, and in the resolution of the resulting database: finer structures aren't well represented. For individual pixels cloud shadows can play a role.

I think that more analysis and discussion is needed here.

 $\implies$  That would indeed be good subjects for further study.

**Discussion of uncertainties:**

• The manuscript uses relative changes of uncertainties in many places. In my opinion, this should be added by absolute changes in error at least in some places. For example, in Table 3, relative changes of the order of 2% in the SCD errors are reported. If I assume an original error of 10%, this would mean that the error now is 9.8%, right? I think that absolute values give a better impression of how large the improvement is.

 $\implies$  Reporting SCD error changes as percentage "(new - old) / old" is done because that is most informative; absolute changes of the SCD error are less interesting. The top panels of Figs. 2 and 3 give a good idea of the absolute SCD error changes for those users interested in this. To accomodate the referee's remark and similar remarks of referee #2, the absolute change of the SCD and the (corrected) SCD error are now listed in Table 3, while the change in VCDstrat is moved to a new table.

 $\implies$  Note that giving SCD error as persentage of the SCD value is not a good idea because: a) The SCD error for the outer 20-odd rows is systematically higher than for the other rows, due to the on-board across-track binning of spectra, as can be seen in the top panel of Fig. 3 and figures in van Geffen et al., 2020.

b) The SCD varies strongly across-track due to the viewing geometry; the GCD is fairly horizontal (bottom panel in new Fig. 3) but giving the SCD error as percentage of the is probably not making things clearer.

• I could not find the definition of the RMS error, which is also given in the text and table. Please add the definition and explain why it makes sense that the RMS error sees larger reductions than the SCD error.

 $\implies$  The definition of the RMS error, along with the DOAS fit model details and the definition of the  $\chi^2$ , is given in the paper Van Geffen et al. (2020) and in the ATBD. To make this clearer the first paragraph of Sect. 3.3 is expanded and for readability sake split in three paragraphs (here indicated by ||):

... orbits of each of the DDS-2 and DDS-3 periods. || The main SCD retrieval results shown here are the SCD value and the associated error following from the DOAS fit, as well as the RMS error: the root-mean-square of the so-called fit residual, i.e. of the difference between the modelled and the measured reflectance, which serves as a measure for the quality of the fit. Another such measure is the magnitude  $\chi^2$ , the chi-squared merit function that is minimised in the DOAS fit, which takes into account the uncertainty on the measured reflectance (the RMS error does not). Definitions and other details of the DOAS fit approach are given by van Geffen et al. (2020) and can also be found in the ATBD (van Geffen et al., 2021). || The averages in Fig. 4 and Table 3 are not an ...

 $\implies$  The SCD error is an estimate for the accuracy of the SCD, while the RMS error follows from the fit residual, so they respond differently to changes in the input (the level-1b spectra).

• One key aspect of the manuscript is the description of changes in the NO2 columns. For users, it is important to know if the algorithm changes lead to NO2 variations within the error bars or outside the error bars. In other words: Do they have to worry that conclusions they drew on old TROPOMI data have to be revised or was this already covered by the uncertainties given in the product?

 $\implies$  The uncertainty in VCDtrop over polluted areas can be larger than the change of VCDtrop there, as you can see from comparing the figure below and the bottom row for Fig. 5. Whether users have to adapt their conclusions depends on how they use the TROPOMI data. For applications that depend on absolute columns (like emission estimates, surface NO2 estimates, validation) have to be re-done using the new column data. Trend analysis, which uses relative columns, need no updated as long as the period over which the analysis takes place does not include a major algorithm change (like the switch to FRESCO-wide). We cannot provide general instructions for all applications, though, but it is likely that conclusions over polluted scenes will change, but not necessarily for all cases, as is also indicated by the validation results in Sect. 5.

Figure 1: ... tropospheric VCD error values of the v2.1 data (left column) and the "DDS minus OFFL" difference (right column), ...

**Comparison to QA4ECV OMI product:**

Comparison to OMI data is important and useful to identify problems. In version v2.2, the agreement between the operational TROPOMI product and the QA4ECV OMI product now is good for the vertical tropospheric columns which is nice. This has been achieved mainly through two changes: 1) the use of the wide FRESCO cloud pressure and 2) the correction of surface reflectance in cases of negative cloud fractions. However, neither of the two corrections is applied in the QA4ECV product. If a similar surface reflectance correction would be implemented in the QA4ECV+ product, differences would increase again.

 $\implies$  The use of the FRESCO-wide cloud pressures is indeed a major improvement in the operational TROPOMI product, as witnessed by comparisons to various other data products (e.g. ground-based NO2 columns, VIIRS cloud parameters) of which the OMI QA4ECV product is one. The impact of the narrow absorption bands on the FRESCO retrieval, leads to cloud pressures that are too close to the surface, which is corrected by the FRESCO-wide approach. This applies to all instruments where FRESCO has been applied, as was shown by Marine Desmons, unfortunately unpublished. Since OMI does not have a NIR channel, the cloud properties are taken in the VIS channel with a different method, hence the "1)" change mentioned simply does not apply to OMI and so the comparison between TROPOMI and OMI remains the same.

For the reader to better understand the difference between the FRESCO versions, in view of the fact that the Eskes et al. (2022) paper is still "in preparation" and addressing comments of the other reviewer, the beginning of Sect. 4.1 is adapted:

The FRESCO+ algorithm (Wang et al, 2008) retrieves cloud information from the  $O_2$  A-band around 758 nm (cloud fraction and cloud pressure) as well as scene parameters assuming clear-sky (scene albedo and scene pressure) and was developed for the GOME-2 instrument. Due to the high spectral resolution of TROPOMI compared to GOME-2, the fact that TROPOMI has a spectral smile (cf. Sect. 3.1), and because of TROPOMI's row-dependent instrument spectral response function (ISRF, known also as slit function) with spectral shifts caused by inhomogenous slit illumination, the FRESCO+ algorithm needed to be re-written and the corresponding lookup tables needed to be generated once more. The resulting implementation is called FRESCO-S (short for FRESCO-Sentinel) and its cloud pressure data is used for the v1.2-v1.3 NO2 data product.

FRESCO+ (Wang et al., 2008) makes use of the wavelength ranges 758 - 759 nm, 760 - 761 nm and 765 - 766 nm. For the FRESCO-S implementation the first window, representing the continuum, was shifted a little to 757 - 758 nm. As a further improvement of the cloud retrieval, nicknamed FRESCO-wide, the third window is extended to 765 - 770 nm in order to include more of the weaker O2 absorption lines. As

a further improvement of the cloud retrieval, nicknamed FRESCO-wide, the third window is extended to 765 - 770 nm in order to include more of the weaker O2 absorption lines. This extention mainly impacts the lower clouds, generally decreasing the cloud pressure in the order of 50 hPa, and is relevant for all instruments where FRESCO has been applied. For high clouds the FRESCO versions deliver very similar cloud heights on average. Further details are given in the ATBD (van Geffen et al, 2021). FRESCO-wide, used as of NO2 v1.4, provides a more realistic ...

 $\implies$  The adaptation of the surface albedo of some individual ground pixels in case of negative cloud fractions following from the cloud retrieval in FRESCO and the NO2 fit window may lead to somewhat larger VCDtrop for those pixels. The O2-O2 code used for the current OMI collection-3 data, which is used for OMI/QA4ECV probably does not contain this surface albedo adaptation – the code is rather old and very complex, so I'm told. It thus is likely that if the O2-O2 cloud algorithm would be updated as well, the difference between OMI/QA4ECV and TROPOMI VCDtrop value may increase a little again for those ground pixels. Obviously this does *not* affect the improvement of TROPOMI w.r.t. ground-based measurements.

To mention these points, this paragraph is added at the end of Sect. 4.4:

It should be noted here that the OMI/QA4ECV processing does not apply the albedo adjustment discussed in Sect. 4.3 on the OMI data, which means that for these cases the difference with TROPOMI data may now be underestimated. This issue does not affect the improvement of TROPOMI data with regard to ground-based measurements. (The forthcoming collection-4 OMI NO2 reprocessing will contain the albedo adjustment algorithm of TROPOMI.)

Similarly, if Figure 7 can be taken as an indication, implementation of the O2-O2 cloud pressure in TROPOMI data would move NO2 vertical columns back towards values seen in v1.2, again increasing differences to OMI. I think that this needs to be acknowledged and discussed.

 $\implies$  As mentioned above, we have removed the O2-O2 curve from Figs. 7 and 8 as it apparently leads to confusion. The O2-O2 data is currently still under investigation. Further discussion in the present paper is therefore not necessary.

**Minor comments**

• The reference Eskes et al., 2021 appears three times in the bibliography for different manuscripts and it is unclear, which of the documents you are referring to in the citations given in the text.

 $\implies$  There are two "Eskes et al, 2021" in the reference list: one is the Product User Manual (PUM) the other one is the "in preparation" paper. Though the PUM was referenced only once (sect. 1.1), this could indeed be confusing. For now this is "solved" by the fact that the "in preparation" paper needs the year 2022, but in case of an update of the PUM we will use "2022a" and "2022b" to be clear. By the way, since "Eskes et al, 2022" is still "in preparation", the number of references to it has been reduced.

 $\implies$  The 3rd reference the reviewer means is "Eskes and Eichmann, 2021", the Product ReadMe File (PRF) and should as such be clear.

• Page 4, line 16: Updated level1b v2.0 – surely, the updated level1 data will have a new version number?

 $\implies$  At the moment of writing the manuscript the discussion which version number the updated L1B, i.e. including radiance degradation correction, will have was ongoing. In Jan. 2022 it was decided that this will be v2.1.0, written as "v2.0" in the manuscript.

• Page 5, line 11: is used  $\Rightarrow$  are used

 $\implies$  You are right – it has been corrected.

• Page 6, line 13: "corrections for the absolute and relative (ir)radiances" this is very vague and also the destinction between absolute and relative (irr)radiaces is unclear – to my knowledge, the lv1 product only contains absolute (irr)radiances. Please expand and clarify.

 $\implies$  We understand that "relative irradiance" can be confusing – it has to do with details in how the level-0 is treated and calibrated. The paragraph in Sect. 2.1.3 is been reformulated:

In the updated irradiance product (Ludewig et al., 2020) the signal is corrected for optical degradation.

In addition, there are improvements in the absolute irradiance calibration and the correction for the solar angle dependence of the irradiance signal. Furthermore, noise and error estimates of the irradiance spectra and the determination of the measurement quality are improved ...

• Page 7, line 4: trace gas concentrations  $\Rightarrow$  trace gas columns

 $\implies$  Yes, that is a better way of phrasing – it has been corrected here and on page 4, line 24.

• Page 7, line 26: wavelengths varies  $\Rightarrow$  wavelength varies

 $\implies$  Thanks for spotting this – it has been corrected.

• Page 7, line 30: If only two corrections are made to the spectral index selection, why does this have any effect on the other rows?

 $\implies$  The two corrections applied are in the general spectral index selection, i.e. they apply to all rows. The new selection clearly improves the SCD fit results for the rows around the row where  $i_b$  changes mentioned in the paper. For other rows the new selection may lead to one or two extra spectral pixels included in the fit than before, which may lead to small differences in the fit results. Hence the phrasing "with little to no effect on other rows".

• Page 8, line 23: Why is it, that some rows always have outliers, and why at the beginning of the NO2 fitting window? Is there a reason for this unusual behaviour? Please expand.

 $\implies$  The observation that in pre-v2.1 data orbits have outliers for most pixels along rows where the start spectral index of radiance and irradiance differ – due to the difference in their respective (nominal) wavelength grids as a result of the (correction for the) Doppler shift in the irradiance – lead us to realise that the spectral pixel selection in the radiance and irradiance for the fit window was not optimal. With the corrections mentioned in Sect. 3.1 this was largely solved: as of v2.1 there are no rows that systematically contain an outlier. The text has been expanded to make this clearer:

 $\dots$  these are rows for which most v1.2 pixels have exactly one outlier at the beginning of the NO2 fit window, while the corresponding v2.1 pixels have not, i.e. these lines  $\dots$

• Figure 2: Please add matching figure for slant column differences

 $\implies$  Done; for completeness sake Fig. 3 is also expanded with the SCD (in terms of the GCD). As requested by referee #2 maps of the SCD error value without and with outlier removal are added as figure to the Appendix.

• Page 13, line 3: I assume that the RMS error is mainly affected by reflectance noise. You probably meant "by the assumed reflectance noise"

 $\implies$  The RMS error follows from the fit residual, but the equation does not contain the reflectance noise itself. But the reflectance noise indeed influences the fit quality through the signal-to-noise ratio (SNR): larger noise means lower SNR, and from that lower fit quality. Note that the reflectance noise is included in the  $\chi^2$  of the DOAS fit. See also the reply to the 2nd bullet of "Discussion of uncertainties" above.

• Page 13, line 12: What do you base the conclusion on, that v2.x data have "much improved DOAS fit quality"? In the table, I see relative reductions of the uncertainty by around 2%, which is an improvement but a rather small one.

 $\implies$  The RMS error is a measure for DOAS fit quality and it decreased by a substantial 7%. Still, perhaps the word "much" is too much – it has been removed.

• Page 13, line 26: There is a clear sea-land contrast visible in the differences. Please discuss.

 $\implies$  It is not clear to which sea-land difference, in which panel, the referee refers. In fact, it may be a little surprising that one sees so little variation in NO2 along the coast, for example where plumes move from land to see, in both versions, even though there is a "jump" in the albedo at the coastline. Perhaps the referee means the bottom-right panel, the difference in the tropospheric VCD? Here a land-sea contrast can be explained easily: the larger NO2 sources are on land and the difference between the two retrievals scales with the NO2 column. To make this clear, the following has been added to Sect. 4.4:

... The bottom right panel of Fig. 5 shows some land-sea contrasts in the tropospheric VCD difference:

sources of large  $NO_2$  concentrations are on land and the difference between the two retrievals, which is chiefly caused by cloud pressure differences, scales with the  $NO_2$  column value.

- Page 14, line 8: lower spatial distribution  $\Rightarrow$  lower spatial resolution
  - $\implies$  You are quite right it has been corrected.
- Page 16. Line 11: not sure how sun glint is related to the change in snow/ice map

 $\implies$  Oops, sorry: the phrase "and cloud from sun glint" has to be removed (the cloud algorithm has some difficulty with clouds over sun glint, but that is not relevant here).

• Page 16, line 16: Please provide some information on how the ECMWF product manages to be better than the NISE product in so many aspects. Is there a reference for how ECMWF does that? Is it using the same input data? Has it been validated?

 $\implies$  In some areas the NISE flag is based on observations once every 3 to 7 days, and as a result snow episodes may be missed, which is particularly relevant for moderate latitudes and coastal areas. The ECMWF snow/ice flag is determined from the parameters for snow depth (sd, 128.141, in meter water equivalent) and snow density (rsn, 128.33, in kg/m3). The source of the information is synoptic data and data from the Interactive Multi-sensor Snow and Ice Mapping System (IMS), which according to Cooper et al. (2018) is favoured over the NISE data for TEMPO trace gas retrievals. The text of Sect. 4.2 is adapted accordingly:

... may be important, while short term snow episodes occuring at mid-latitudes may be missed due to the avering over multiple days. As of v2.1... the NO2 data. The ECMWF snow data (De Rosnay et al., 2015) is derived from synoptic data and from the Interactive Multi-sensor Snow and Ice Mapping System (IMS); Cooper et al. (2018) show that IMS has better agreement with in situ observations over North America and that NISE misses a significant number of snow-covered pixels. ...

• Page 17, line 6: "NO2 surface albedo is adjusted from the value of 0.62 in the climatology to a more realistic 0.04" I guess what you meant to say is, that because of the improved NISE, the normal albedo is used instead of the value appropriate for sea ice.

 $\implies$  Not exactly: adaptation of the climatological surface albedo takes place only if no sea-ice is reported; if non-zero sea-ice is reported the climatological value is used as-is. To make this clearer the end of Sect. 4.2 is expanded a little:

 $\dots$  adjusted from the value of 0.62 in the climatology to a more realistic 0.04. Such adjustments are made only in for cases without any snow or ice reported.

• Page 26, line 24: "The NO2 cloud (radiance) fraction is currently derived from the FRESCO cloud pressure". This sentence does not make sense to me, please check.

 $\implies$  The statement is correct. The cloud (radiance) fraction in the NO2 fit window is derived from cloud pressure, surface pressure, surface albedo and cloud albedo, where the latter two are the values for 440 nm, i.e. for the NO2 fit window, using a look-up table (LUT) generated with an RTM. The cloud pressure comes from the cloud product, which is currently FRESCO. To make this cleared "derived from" is changed into "using" and Sect. 4.1 contains a mention of the use of a LUT in a newly added last paragraph.

• Page 27, line 10: data data

 $\implies$  Thanks for spotting this – it has been corrected.

- Page 28, line 9: See above
  - $\implies$  See answer above.
- Page 28, line 19: "The use of the improved level-1b v2.0 leads a) to a somewhat lower cloud pressure for" I somehow fail to see where this has been shown in the paper

 $\implies$  Sect. 4.1, Fig. 7.

- Page 28, line 21: number fully  $\Rightarrow$  number of fully
  - $\implies$  Thanks for spotting this it has been corrected.

---

## Author Comment (AC2)

**Reply to Comment on amt-2021-329 by Anonymous Referee #3**
* * *
Referee comment on "Sentinel-5P TROPOMI NO2 retrieval: impact of version v2.2 improvements and comparisons with OMI and ground-based data" by Jos van Geffen et al., Atmos. Meas. Tech. Discuss., https://doi.org/10.5194/amt-2021-329-RC2, 2021
* * *
$\Longrightarrow$ The referee report is copied below; the reply is preceeded by an arrow, like this text.

The paper "Sentinel-5P TROPOMI NO2 retrieval: impact of version v2.2 improvements and comparisons with OMI and ground-based data" by Geffen et al. presents the improvements made in the official TROPOMI NO2 product. It is well written, matches the scope of AMT, and will serve as a key reference for the TROPOMI NO2 data set which is frequently used by the community.

$\Longrightarrow$ We thank the referee for these kind words.

I recommend publication after dealing with some minor comments.

**Minor comments**

- Page 2 line 2: I understood that the main effect is on the tropospheric column. I would skip "and total" here.

  $\Longrightarrow$ As concluded in Sect. 5, the situation for individual measurents may be quite complex, both for tropospheric and total columns – hence this phrasing.

- Page 2 line 21: "over e.g.". I think that there are more similar products, especially over the US.

  $\Longrightarrow$ We are not aware of regional data products over the US based on TROPOMI data. There is Laughner et al, 2018 (doi:10.5194/essd-2018-66) but that uses OMI data. We are told NASA is working on a TROPOMI version, but details are not known.

- Page 3 line 2: As this often leads to confusion, it might be worth pointing out that the AMF depend on the profile *shape*, but not on the absolute concentration levels.

  $\Longrightarrow$ You are right and it is indeed good to emphasize this:

  . . . pressure, the shape of the $NO_2$ vertical profile (not of its absolute concentration levels), and . . .

- Page 3 line 5: "v1.2/v1.3": It would be helpful if the authors would add a reference to a table listing the different NO2 product versions and what is new in 1.3/1.4/2.1.

  $\Longrightarrow$ This comes in Sect. 1.1; a forward reference to that is added to the end of the preceeding paragraph:

  . . . ground pixel in question. An overview of the $NO_2$ data versions is given below in Sect. 1.1.

- Page 3 line 20: I appreciate that the authors explicitly provide this scaling factor. In addition, I would appreciate if the unit molec/cm2 would be added at the top of all colorbars or at additional top/right axis.

  $\Longrightarrow$ The molec/cm2 unit is available on the right axis of Fig. 3. That unit has also been added at some locations (Sects. 3.1, 3.2, 3.3, 4.3, 4.4) and to the upper axis of scatter plots (Figs. 6, 11, 12, 13; addition also to the right axis is unpractical). Adding that unit to the maps (Fig. 2, 5), which are made using Panoply, in the form of a double colour bar is unfortunately not feasible.

- Page 3 line 26: "and include information on earlier versions." I don't understand this: does this mean that the linked document includes this information? (then add an "s"). Or is this a reminder to the author to include this information here?

  $\Longrightarrow$ That should have been "and includes information" – thanks for spotting this.

- Page 3 line 28: Please name the institutes directly. Pointing out the country here reads like strange kind of nationalism.

⟹ The country was given because of the "NL" in the processor name. But you are right: this formulation doesn't look good – it has been corrected:

... the so-called NLL2DP ("Netherlands level-2 data processor") that provides the TROPOMI data products for which KNMI and SRON are responsible, ...

- Page 4: Here comes the information I was asking for above. I still think it would be helpful to have a table with the information in short version, even if this would add some redundancy.

  ⟹ The itemized list of the different versions contains all necessary information and is, we believe, easier to read than a table. Adding this also in the form of a table would be an exact doublure, as there is no clear way to summarise the already short descriptions.

- Page 4 line 16: I have seen TROPOMI results before 30 April 2018, and as far as I am aware the quality of TROPOMI spectra was already quite good since beginning of 2018. I would highly appreciate if the reprocessing would be extended back to 1 Jan 2018.

  ⟹ Several things are important here.
  a) 30 April 2018 was the beginning of the so-called operational phase, under the responsibility of ESA, who is also responsible for the reprocessing, and 30 April 2018 thus marks the start of the official data product release.
  b) There is indeed data before that, but during this so-called commissioning phase, especially up to early March there are large gaps in the data (when instruments tests and special measurements were done). Regular data is available only since 15 March 2018.
  c) For the $NO_2$ data assimilation with TM5-MP to provide reliable results, a spin-up periods of several days is required, hence data prior to mid March 2018 would not be of optimal quality if it was processed. Note that because of the spin-up, the actual reprocessing will start on 15 April 2018, with data released as of 30 April.

- Section 3.2: A similar outlier removal was introduced by Richter et al., https://doi.org/10.5194/amt-4-1147-2011. Please add a reference here.

  ⟹ Not quite "similar": the outlier removal we use sets limits based on the inter-quartile range and does this only once (that is: after the first round of the removal, there is no second round), while Richter et al. 2011 "itteratively scan the residual for points having a value larger than 10 times the average residual of the fit. [...] This procedure is repeated until no further outliers are identified." Since there are other ways to do outlier removal and a discussion of these approaches is beyond the scope of the paper, adding this one reference does not seem appropriate.

- Fig. 2: The figure illustrates that the outlier removal reduces the error, but the numbers are out of context. Instead (or in addition) of showing the difference I would propose to show the maps of SCD errors with and without outlier removel directly.

  ⟹ The main idea of the plot is to show that the SCD error decreases (negative values on the map) and where (mainly in the SAA). What the absolute SCD error values are is of less importance, but to give some idea average SCD error are shown in Fig. 3. To nevertheless accomodate the referee's request, maps of the SCD error values with and without outlier removal are added as figure in the Appendix. Note that referee # asked for a map of the SCD value difference, and this was added to Fig. 2; for completeness sake Fig. 3 was then also expanded with the SCD (in terms of the GCD).

- Table 3: Again, it is difficult to assess what a relative change of SCD error of 1% means, if the SCD error itself is not given. I would propose to compare the SCD error with and without outlier removal etc. directly.

  ⟹ Comparing the SCD error with and without outlier removal is done in Figs. 2 and 3 for one example orbit, with the v2.2 processor, i.e. no other differences than the outlier removal. This is dedicated processing and therefore not available for all periods. The difference between DDS and OFFL contains more than just the outlier removal, showing the total effect of all improvements is in Fig. 4 and Table 3. To accomodate the referee's remark and similar remarks of referee #1, the absolute change of the SCD and the (corrected) SCD error are now listed in Table 3, while the change in VCDstrat is moved to a new table.

  Please add a footnote why the last entry for Vstrat is missing.

⟹ With the VCDstrat data moved to a new table, this is no longer necessary.

- Page 11 line 11: "strong decrease in the SCD error and RMS error": I would like to have this "strong" decrease given in absolute numbers in the presented figures and tables.

  ⟹ The absolute SCD error changes are now listed in Table 3 (and an example can been seen in the new Fig. A1). Absolute changes in the RMS are not listed because the absolute RMS cannot be compared to other retrieval approaches due to differences in the RMS definition. (In addition, Table 3 currently fits just in one fullpagewidth.)

- Page 14 line 15: Please explain why a dedicated TROPOMI version for FRESCO was necessary and FRESCO+ could not be used directly.

  ⟹ In answer to this and the following two referee comments.
  Unfortunately there is no official FRESCO-S or FRESCO-wide publication (paper or published report) as yet. The idea was here to refer to the discussion in the Eskes et al. paper that is "in preparation". Since that paper is still in preparation and in view of this referee's comments, the number of references to that paper is reduced and it was decided to adapt the beginning of Sect. 4.1:

  The FRESCO+ algorithm (Wang et al, 2008) retrieves cloud information from the $O_2$ A-band around 758 nm (cloud fraction and cloud pressure) as well as scene parameters assuming clear-sky (scene albedo and scene pressure) and was developed for the GOME-2 instrument. Due to the high spectral resolution of TROPOMI compared to GOME-2, the fact that TROPOMI has a spectral smile (cf. Sect. 3.1), and because of TROPOMI's row-dependent instrument spectral response function (ISRF, known also as slit function) with spectral shifts caused by inhomogenous slit illumination, the FRESCO+ algorithm needed to be re-written and the corresponding lookup tables needed to be generated once more. The resulting implementation is called FRESCO-S (short for FRESCO-Sentinel) and its cloud pressure data is used for the v1.2-v1.3 $NO_2$ data product.
  . . .
  FRESCO+ (Wang et al., 2008) makes use of the wavelength ranges $758 - 759$ nm, $760 - 761$ nm and $765 - 766$ nm. For the FRESCO-S implementation the first window, representing the continuum, was shifted a little to $757 - 758$ nm. As a further improvement of the cloud retrieval, nicknamed FRESCO-wide, the third window is extended to $765 - 770$ nm in order to include more of the weaker $O_2$ absorption lines. This extention mainly impacts the lower clouds, generally decreasing the cloud pressure in the order of 50 hPa, and is relevant for all instruments where FRESCO has been applied. For high clouds the FRESCO versions deliver very similar cloud heights on average. Further details are given in the ATBD (van Geffen et al, 2021).
  FRESCO-wide, used as of $NO_2$ v1.4, provides a more realistic . . .

- Page 14 line 18: Does "FRESCO" here mean (a) all FRESCO versions or (b) the original FRESCO version, different from FRESCO+ and FRESCO-S?

  ⟹ FRESCO here refers to the general FRESCO approach, not a specific version; the text makes this clearer:

  . . . (which the FRESCO algorithm sees as an effective cloud) . . .

- Page 15 line 3: Please add a reference to "FRESCO-wide", or provide more detail here.

  ⟹ See above.

- Figure 9: With the chosen color bar, it is impossible to discriminate between 100I propose to use a colobar which is monochromatic from 0 to 100 (e.g. dark blue to light blue) and have additinal discrete and distinctive colors for the discrete cases occuring above 100.

  ⟹ The point of the maps and the text in Sect. 4.2 is that the quality of the ECMWF data is much better than the NISE data in resolution and an particular in coastal areas, and that the ECMWF does not contain the problematic flags 252-254 of NISE. It is indeed difficult to see the difference between the flags 100, 101 and 103, but that distinction is not relevant for the $NO_2$ data: when in the snow/ice mode – which is when the snow/ice flag is 003 larger (but not 255) – the amount or type of snow/ice is not used. As additional information to the referee, the maps below zoom in on the flage 100-103. To better reflect the usage in the $NO_2$ processing, the following sentence is added to the end of the second paragraph of Sect. 4.2:

  . . . the problematic NISE flags 252-254. The cloud fraction and cloud pressure are used for the AMF calculation for pixels flagged as ocean (255), snow-free land (000) or a percentage sea-ice flag smaller than for the ECMWF data (in case of NISE data this was smaller than 002, since NISE has the

tendency to underestimate snow/ice cover); in case of other flags the scene parameters are used.

[Figure]

Figure 1: Maps of Fig. 9 of the manuscript focussing on the flags 100 (100% sea-ice), 101 (permanent ice) and 103 (snow) in the NISE (left) and ECMWF (right) snow/ice cover data.

- Figure 13: "version" should be "versus"?

  ⟹ Thanks for spotting this – it has been corrected.

- Figure 15: The meaning of IP68 has to be explained in the caption.

  ⟹ Addition to the figure caption, at the and of (a):

  ... (dotted lines). IP68 is the central 68 interpercentile range, the difference between the 84th and 16th percentiles, a measure for the dispersion. $\Delta$ is the difference "S5P-GB", where GB stands for ground-based.

  ⟹ And the text near the end of the first paragraph of Sect. 5.1 is expanded:

  ... and the dispersion (half of the central 68 interpercentile, shorthand 0.5 IP68) of the difference between ground-based ("GB") and S5P ...

- Page 29 lines 12-17: Please check if you could be more concrete by now on these future plans.

  ⟹ The text in Sects. 1.1, 6.x and here have been adapted to reflect that NO2 v2.4 and the subsequent mission reprocessing will contain i) a new version level-1b spectra with radiance degradation correction, and ii) the new TROPOMI surface albedo climatology.

---

## Author Response (AR2)

Dear Lok

Many thanks for your kind words and for reading the manuscript carefully.

With "App." is meant "Appendix" (akin to "Ch." and "Sect.") -- for clarity
I've replaced it with the full word.

And I've corrected the two typos.

Best regards,

Jos.